# Proposer-Agent-Evaluator (PAE):
# Autonomous Skill Discovery for Foundation Model Internet Agents

**Yifei Zhou** [* 1]  **Qianlan Yang** [* 2]  **Kaixiang Lin** [3]  **Min Bai** [3]  **Xiong Zhou** [3]  **Yu-Xiong Wang** [2]  **Sergey Levine** [1]
**Erran Li** [3]

## Abstract

A generalist foundation model agent needs to have a large and diverse skill repertoire, such as finding directions between two travel locations and buying specific items from the Internet. If each skill needs to be specified manually through a fixed set of human-annotated instructions, the agent's skill repertoire will necessarily be limited due to the scalability of human-annotated instructions. In this work, we address this challenge by proposing Proposer-Agent-Evaluator (PAE), an effective learning system that enables foundation model agents to autonomously discover and practice skills in the wild. After a context-aware task proposer generates instructions based on website information, the agent policy attempts those tasks in the real world with resulting trajectories evaluated by an autonomous VLM-based success evaluator. The success evaluation serves as the reward signal for the agent to refine its policies through RL. We validate PAE on challenging vision-based web navigation, using both real-world and self-hosted websites from WebVoyager and WebArena. Our results show that PAE significantly improves the zero-shot generalization capability of VLM Internet agents (around 50% relative improvement) to both unseen tasks and websites.

## 1. Introduction

The vision of a broadly capable and goal-directed agent has long captured the imagination: from agents that can browse the web to fulfill user instructions to robots that could perform diverse tasks in the home, AI systems that can independently accomplish open-world goals would not only present tremendous practical value, but their development would also elucidate some of the deepest questions in AI. Recent advancements in foundation models provide us with a compelling possibility for how this vision could be realized (OpenAI, 2024; GeminiTeam, 2024). Pre-trained foundation models have been used to explore generalist agents (Liu et al., 2023b) in real-world decision-making scenarios, such as navigating websites to make travel plans (He et al., 2024a) and solving real-world Github issues (Jimenez et al., 2024). To succeed in these decision-making domains, goal-directed post-training is often needed to elicit long-horizon reward-maximizing behaviors such as information seeking (Hong et al., 2023a) and recovery from mistakes (Bai et al., 2024), instead of only imitating the most probable actions in the pre-training corpus.

A crucial requirement for a successful post-training approach is to endow the generalist agent with a large and diverse goal-directed skill repertoire. This can include finding directions between two travel locations and buying specific items from the Internet, which the agent can then exploit to solve real-world tasks proposed by users. However, manually specifying the skills (Deng et al., 2023) (i.e. through a static set of human-annotated instruction templates such as "Find the driving directions and estimated time to travel from Location A to Location B") will likely result in a limited skill repertoire. In particular, generating high-quality human-annotated task templates can be expensive, making it impractical to scale up. The use of a small set of task templates fails to capture the range of skills an agent needs for the full breadth of the real world, leading to distribution shift problems when deployed at the test time. Therefore, it naturally raises the research question: *instead of requiring users to manually define tasks for foundation model agents, can these agents **autonomously discover** and practice potentially useful skills on their own?*

To answer this question, we first note the observation of the **asymmetric capabilities of SOTA VLMs as skill proposers/evaluators and as agents** (ablation experiment results in Section 5) in many realistic task settings such as web agents. Intuitively, VLMs are very good at conceiving common tasks that people would like to perform on a shopping website (e.g. adding a specific product to cart) and confirming

---

[*]Equal contribution   [1]University of California, Berkeley  [2]University of Illinois, Urbana-Champaign  [3]Amazon. Correspondence to: Yifei Zhou <yifei_zhou@berkeley.edu>.

*Proceedings of the 42nd International Conference on Machine Learning*, Vancouver, Canada. PMLR 267, 2025. Copyright 2025 by the author(s).

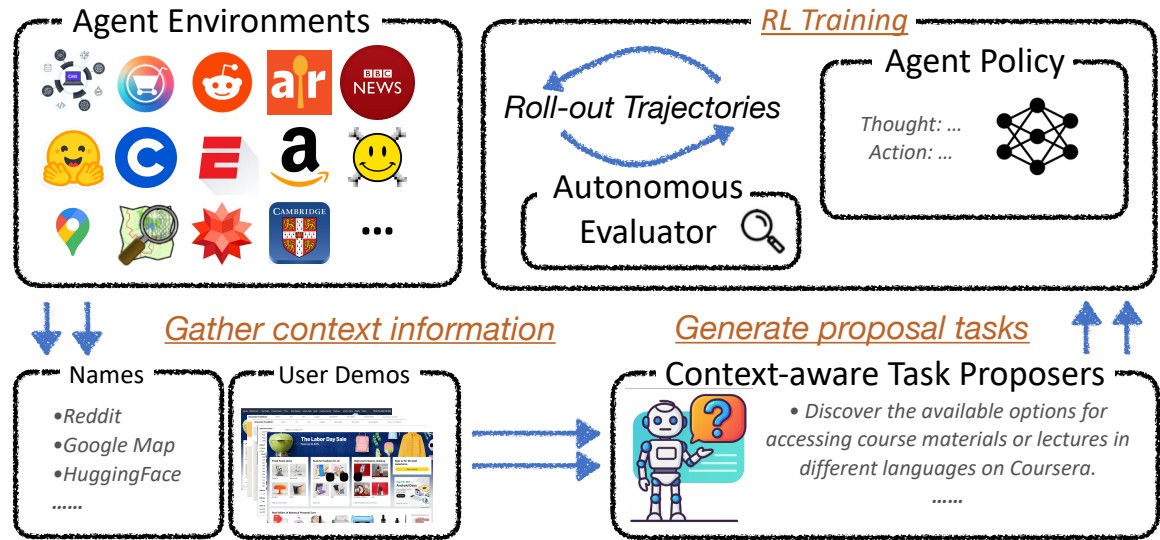

*Figure 1.* **An overview of Proposer-Agent-Evaluator** showing the main components of our autonomous skill discovery framework, that endows the agent with autonomously discovered skills for future human requests.

whether a specific product has been added to the shopping cart (e.g. by looking at the final screenshot to see if the shopping cart contains the product), while worse at actually navigating through the web to find the product and add it to the cart. If our system can be designed to maximally exploit this asymmetric gap, we may bootstrap the performance of the agent capability by reliable signals from task proposers and evaluators. However, merely discovering possible skills is not enough, since the ultimate goal is to do well for unseen real-world human requests. The right design choices of the system also need to be carefully validated to ensure that the autonomously discovered skills are actually useful and can generalize well to unseen human requests.

In this work, **our main contribution** is to propose an effective learning system, Proposer-Agent-Evaluator(PAE), for agents (in particular, Internet agents) based on foundation models to autonomously discover new skills without any human supervision and these new skills can be effectively exploited to solve unseen real-world human-annotated tasks in a zero-shot manner. At a high level, PAE identifies the interesting skills with a task proposer, attempts them with an agent policy, and performs an online RL loop based on the reward provided by an autonomous evaluator. PAE identifies the following design choices to obtain reliable signals from the task proposer and evaluator, and ensure that the agent can extrapolate from autonomously discovered to unseen human requests. First of all, to propose feasible and realistic tasks, PAE employs a class of context-aware task proposers where the context of functions and constraints crucially define what actions are supported by the specific environments (e.g., creating a reddit post) while others may not be supported (e.g., checking the protected information of other users). Such context can be implicitly defined from

different sources and are shown to be effective, such as user demos and even website name alone! Moreover, to obtain the most robust reward signal without accessing the hidden state information, we apply an image-based evaluator that only provides sparse 0/1 rewards based on the final outcome. Finally, we design an additional reasoning step before the agents output actual actions, which enables the agents to better reflect on their skills and results in a significant improvement in its generalization to unseen tasks.

The scope of our experiments covers challenging end-to-end vision-based web navigation, where the observation space simply contains the screenshot of the current web page and the action space contains primitive web operations such as clicking on links and typing into text boxes. We validate the effectiveness of PAE framework with realistic web-navigation benchmarks, including more than 100 domains both from online websites like Amazon from WebVoyager (He et al., 2024a) and self-hosted websites like PostMill from WebArena (Zhou et al., 2024a). In our experiments, we find that PAE with LLaVa-1.6 (Liu et al., 2024) as the agent policy can autonomously discover useful skills through interactions with various websites without any human supervisions, even when the task proposer and evaluator model is significantly worse than the agent model. More importantly, our results demonstrate that these skills can zero-shot transfer to unseen test instructions and even unseen test websites. On websites from WebVoyager and WebArena, PAE attains around 50% relative improvement in success rate, enabling LLaVa-1.6-7B to achieve performance comparable with LLaVa-1.6-34B fine-tuned with demonstration data and Qwen2VL-72B (Yang et al., 2024a).

## 2. Related Works

**Foundation model agents.** Thanks to the generalization capabilities of Large Language Models (LLMs) (Brown et al., 2020; Llama3Team, 2024; GeminiTeam, 2024) and Vision Language Models (VLMs) (OpenAI, 2024; Liu et al., 2024; Wang et al., 2024c; Liu et al., 2023a; Zhai et al., 2024), recent works have successfully extended such agents to more general real-world use cases (Bai et al., 2024; Zheng et al., 2024; He et al., 2024a; Zhang et al., 2023a; Zhou et al., 2024a; Koh et al., 2024; Gur et al., 2021; Furuta et al., 2024; Lin et al., 2024; Xu et al., 2024a;b). Besides constructing prompting wrappers around proprietary VLMs (Zhang et al., 2023a; He et al., 2024a; Zheng et al., 2024; Xie et al., 2024; Yang et al., 2024b; Wang et al., 2023a) and fine-tuning open-source VLMs with expert demonstrations (Gur et al., 2021; Hong et al., 2023b; Furuta et al., 2024; Zhang & Zhang, 2024; Zeng et al., 2023; Chen et al., 2023), a recent trend has emerged involving interactive improvement of LLM/VLM, in particular web/GUI agents, through autonomous evaluator feedback (Pan et al., 2024; Bai et al., 2024; Putta et al., 2024; Wang et al., 2024b), where evaluator LLMs/VLMs are prompted to evaluate the success of the agents to serve as the reward signal. This approach aims to elicit goal-oriented and reward-optimizing behaviors from foundation models with minimal human supervision. However, these methods still depend on a static set of human-curated task templates, constraining their potential and scalability. Our work introduces a framework where agents can *discover* and practice the skills they find useful, thereby eliminating the reliance on predefined and human-curated task templates and opening up new possibilities for scalability and adaptability in training generalist autonomous LLM/VLM agents.

**Self-generated instructions.** Self-generated instructions for improving LLMs have been shown to be effective in single-turn LLM alignment (Wang et al., 2023b; Yuan et al., 2024; Wu et al., 2024; Wang et al., 2024a) and reasoning (Pang et al., 2024) domains without interactions with an external environment. AgentGen (Hu et al., 2024) takes a step further to fine-tune LLM agents with expert trajectories in self-generated environments and tasks. However, its feasibility in the self-play agent setting with RL and autonomous evaluators has not been understood. On the other hand, the closest works to ours employ autonomous RL and foundation model task proposers to simplified environments such as games (Zhang et al., 2024a; Faldor et al., 2024; Colas et al., 2023; 2020) and robotics settings with limited number of scenes (Du et al., 2023; Zhang et al., 2023b; Zhou et al., 2024c). While they have shown that the use of autonomous RL combined with foundation model task proposers can help the agent learn diverse skills, this work takes an important step forward to study when those skills can generalize to human requests in realistic benchmarks in the context of web agents and what the best design

choices are. While some concurrent works (Qi et al., 2024; He et al., 2024b; Murty et al., 2024) also apply autonomous task proposals to training web agents where instructions and evaluation rewards are generated from stronger proprietary models, our PAE system focuses on exploiting the asymmetric capabilities of foundation model web agents to achieve self-improvements, where even weaker models can be used to improve the performance of stronger agents.

**Unsupervised skill discovery in deep RL.** Unsupervised skill discovery has been an important research direction in the field of traditional deep RL literatrue (Achiam et al., 2018; Eysenbach et al., 2018) where various algorithms have been developed to discover new robotic skills such as humanoid walking without the need of explicitly defined reward functions. Common algorithms in this field aim to discover *every possible* skill (both meaningful skills like walking and less meaningful ones like random twisting) through either maximizing the mutual information between different states and skill latent vectors (Campos et al., 2020; Laskin et al., 2022; Sharma et al., 2020), or maximizing the divergence of each skill as measured in a metric space (Park et al., 2022; 2023; 2024). In contrast, our work *only discovers meaningful skills* as specified through natural language instructions with the help of pre-trained foundation models, significantly reducing the search space of skills in LLM/VLM agent applications with complex state spaces.

## 3. Proposer-Agent-Evaluator (PAE): Autonomous Skill Discovery system For Foundation Model Agents

Next, we will explain the technical contributions of this paper. In this section, we will define the general system of PAE including a task proposer, an agent policy, and an autonomous evaluator. We will begin by formalizing the learning goal of this system and then detail the roles of each key component. In the section to follow, we will explain how we apply PAE to VLM Internet agents.

**Problem setup** We begin by formalizing the problem setup of autonomous skill discovery for real-world agents. The learning goal of PAE is to find a reward-maximizing policy $\pi$ parameterized by $\theta$ in a contextual Markov Decision Process (MDP) environment defined by $\mathcal{M} = \{\mathcal{S}, \mathcal{A}, \mathcal{T}, \mathcal{R}, H, \mathcal{C}\}$, where $\mathcal{S}, \mathcal{A}$ are the state space and action space respectively, and $H$ is the horizon within which the agent must complete the task. We assume that the agent has access to the environment and can collect online roll-out trajectories through accessing the dynamics model $\mathcal{T}$ as a function of determining the next states given the current states and actions. *Crucially, we assume that the ground-truth task distribution $\mathcal{C}$ and the reward function $\mathcal{R}$ are hidden during training and we have to use a proxy task distribution $\hat{\mathcal{C}}$ and reward function $\hat{\mathcal{R}}$ instead.* Consider the

setting of training a real-world Internet agent. The dynamics model $\mathcal{T}$ would be a simulated browser environment that the Internet agent can interact with. The ground-truth task distribution $\mathcal{C}$ might be the distribution of tasks that would be asked by the real users when the Internet agent is deployed and a possible choice for the reward function $\mathcal{R}$ might be whether the agent has satisfactorily completed the tasks for the real users. In such a real-world setting, although the agent can freely access resources from the Internet through a simulated browser environment during training, assuming knowledge of the ground-truth task distribution and reward function is impractical. Therefore, we employ VLM-based task proposers $\hat{\mathcal{C}}$ and reward model $\hat{\mathcal{R}}$ as proxies. The desired outcome is that improving the policy $\pi_\theta$ with $\hat{\mathcal{C}}$ and $\hat{\mathcal{R}}$ can lead to an improved policy **that can successfully generalize to the ground-truth task distribution and reward functions** which are only used as evaluations.

**Key components** Figure 1 shows the interplay between the key components in our framework, including a context-aware task proposer, an agent policy, and an autonomous evaluator. The role of the **task proposer** $\hat{\mathcal{C}}$ is to serve as a proxy to improve on the ground-truth task distribution $\mathcal{C}$ during the learning process. However, it might be unrealistic to expect the task proposer to generate feasible tasks without knowledge of the environment. To provide more context of the functions and constraints of the environment, we assume access to some key information of the environment $z_\mathcal{M}$ based on which the tasks $\hat{\mathcal{C}}(z_\mathcal{M})$ are proposed. In the Internet agent example, this key information can be screenshots of the websites from user demos, or even just the name of the website itself if it is a well-known website such as Amazon.com. Similarly, the **autonomous evaluator** $\hat{\mathcal{R}}$ serves as a proxy of the ground-truth reward function $\mathcal{R}$. The input to the autonomous evaluator is the current state, the current action from the agent policy, and current task that the agent is attempting. In principle, any RL algorithm can be used to update the **agent policy** $\pi$ using a dataset $\mathcal{D}$ that stores all the autonomous interaction data. In practice, we instantiate VLM-based task proposers and autonomous evaluators by prompting foundation models and they are kept unchanged throughout training.

# 4. PAE for VLM Internet Agents

With the general framework set up, we are now ready to discuss the concrete instantiation of PAE in the setting of VLM Internet agents. We start by introducing the environment of vision-based web navigation and then explain how we implement the key components from PAE in this setting.

## 4.1. Vision-Based Web Browsing Environment

We consider the general vision-based web browsing environment (He et al., 2024a; Koh et al., 2024). The goal for VLM agents in this environment is to navigate through realistic web pages to complete some user tasks $c_t$ such as "Inves-

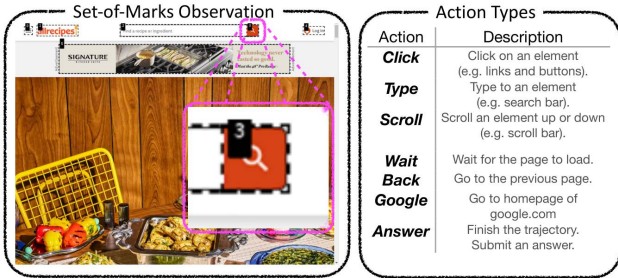

*Figure 2.* **An illustration of the observation space and action space of our vision-based web navigation environment.**

tigate in the Hugging Face documentation how to utilize the 'Trainer' API for training a model on a custom dataset, and note the configurable parameters of the Trainer class". As illustrated in Figure 2, each **observation** $s_t$ from the observation space contains only the screenshot of the last web page just like how humans interact with the Internet. To provide better action grounding, we follow the practice from prior works (Zheng et al., 2024; He et al., 2024a) to augment the observation space with number marks on top of each interactive element such as web links and text boxes. To execute a web browsing action, the Internet agent can directly output the number of the element to interact with and the corresponding action such as clicking and typing, without the need of locating the coordinates of each web element. Therefore each web **action** $a_t$ contains the type of the action to perform and the number of the element to interact with. Each episode finishes either when the agent chooses to finish through the "Answer" action or when a maximum number of 10 steps have been reached. In our experiments, we use ground-truth success detectors (based on either human annotations or functional verifiers) and human annotated tasks from WebArena (Zhou et al., 2024a) and WebVoyager (He et al., 2024a) to evaluate the performance of different policies. Crucially, both the ground-truth success detector and the distribution of human tasks are kept hidden, which challenges the generalization capability of the learnt skills to generalize to a hidden reward function and task distributions.

## 4.2. Context-Aware Task Proposer

In order to generate a diverse set of feasible tasks, we frame task proposing $\hat{\mathcal{C}}$ as a conditional auto-regressive generation based on the context information of the websites. Thanks to the vast pre-training knowledge of relevant context for popular websites like Amazon.com, we find it suffice to use only **website name** as $z_\mathcal{M}$. However, for less common or access restricted websites such as self-hosted websites in WebArena, it is necessary to supply the task proposer with richer context. In the cases of **user demos** being available, we consider an alternative to sample some additional screenshots from the user demos to serve as the context information. In our experiments, we consider both using proprietary models such as Claude-3-Sonnet (Anthropic,

2024) and open-source models such as Qwen2VL-7B (Yang et al., 2024a) for the task proposers, with prompts in Appendix D.

## 4.3. Image-Based Outcome Evaluator

To take full advantage of the asymmetric capability of SOTA VLMs as agents and as evaluators (experimental evidence presented in Section 5), we find it most robust for the autonomous evaluators to complete the easiest evaluation: evaluating the success of the final outcome (Bai et al., 2024; He et al., 2024a) based on the final three screenshots and the agents' final answers to provide only 0/1 response in the end. Other alternatives such as code-based (Zhang et al., 2024a) or step-based evaluations (Pan et al., 2024) are either impractical without access to hidden state information or too noisy because of the hallucination issues present even in SOTA VLMs. In our experiments, we also consider both using proprietary models such as Claude-3-Sonnet (Anthropic, 2024) and open-source models such as Qwen2VL-7B (Yang et al., 2024a), with prompts presented in Appendix D.

## 4.4. Chain-of-Thought Agent Policy

Crucially, as the ultimate goal for the agent policy is to complete human requests, the agent should not only learn diverse skills on the proposed tasks but also reflect on the skills learnt so that they can be helpful for unseen human requests. Therefore, we incorporate an additional reasoning step to outputs the agent's chain-of-thought before the actual web operation. This reasoning step is optimized with the any online RL algorithm just like the actual web operation. For its simplicity and effectiveness, we employ use the REINFORCE objective (Williams, 2004) to optimize the policy with the following loss:

$$\mathcal{L}(\pi) = \mathbb{E}_\tau - (\sum_{i=1}^{H} r(s_i, a_i))(\sum_{i=1}^{H} \log \pi(a_i|s_i)),$$

where $\tau$ is an on-policy trajectory with $H$ steps, and $r(s_i, a_i)$ is the reward at $i$-th step with the action $a_i$ at the state $s_i$. Because the 0/1 terminal reward structure, in this work it turns out to be equivalent to the most simple online policy optimization algorithm Filtered Behavior Cloning (Filtered BC) that simply imitates all thoughts and actions in successful trajectories with the negative log-liklihood loss. This simple method has been widely adopted in prior literature of RL+LLM such as (Zhou et al., 2024b; Snell et al., 2023). This simple policy optimization objective can already lead to a superior generalization capability of the learnt agent. In our experiments, our agent policy is initialized from LLaVa-1.6-Mistral-7B and LLaVa-1.6-Yi-34B (Liu et al., 2024).

## 5. Experiments

The goal of our experiments is to understand the effectiveness of PAE to complete real-world visual web tasks. Specifically, we design experiments to answer the following ques-

tions: **(1)** Can our autonomous skill discovery framework successfully discover skills useful for zero-shot transfer to tasks from an evaluation task distribution unseen to the task proposer? **(2)** How does the models trained with PAE compare with other open-source VLM agents? **(3)** How does the effectiveness of PAE scale with the size and performance of the base model? **(4)** Whether the effectiveness of PAE is limited by the performance of the task proposer and evaluator model?**(5)** How does the use of different contexts (e.g. website names and user demos) affect the performance?

### 5.1. Environments

**WebVoyager** (He et al., 2024a) contains a set of 643 tasks spanning 15 websites in the real world such as ESPN and Arxiv. As tasks in Google Flights and Booking domain are no longer feasible due to website updates, we use the subset of 557 tasks spanning the other 13 websites. Human annotations are carried out for evaluating the success of each trajectory as the ground-truth performance measure.

**WebArena** (Zhou et al., 2024a) is a sand-boxed environment that kept an archived version of 5 popular websites from different domains, including OpenStreetMap, GitLab, PostMill, a store content management system (CMS), and an E-commerce website (OneStopMarket). It includes in total 812 hand-written tasks with functional verifications as the ground-truth reward function. Since GitLab and CMU do not support multi-thread data collection necessary for RL fine-tuning, our experiments are carried out using the task subsets on OpenStreetMap, PostMill, and OneStopMarket. As open-source VLM agents fail to achieve non-trivial performances on PostMill and OneStopMarket (Zhou et al., 2024a), we hand rewrote tasks in those two websites and supplement them with verification functions. Due to these practical constraints, the resulting **WebArena Easy** contains 108 original tasks on OpenStreetMap and 50 rewritten tasks on PostMill and OneStopMarket each.

### 5.2. Baseline Comparisons

We validate the effectiveness of PAE by comparing it with **(1)** proprietary VLMs, **(2)** state-of-the-art open-source VLMs, and **(3)** an alterative supervised fine-tuning (SFT) approach. We consider **Claude 3 Sonnet** and **Claude 3.5 Sonnet** (Anthropic, 2024) for proprietary VLMs, and **Qwen2VL-7B**, **Qwen2VL-72B** (Yang et al., 2024a), **InternVL-2.5-XComposer-7B** (Zhang et al., 2024b), and **LLaVa-Next-7B/34B** (Liu et al., 2024) for SOTA open-source VLMs. All models are prompted similar to He et al. (2024a) using set-of-marks augmented screenshot observations and including chain-of-thought in the action outputs. The prompts are included in Appendix D. As SOTA open-source models struggle to achieve non-trivial performance in the challenging web navigation benchmarks except the largest Qwen2VL-72B, we include another baseline **LLaVa-SFT** that fine-tunes LLaVa with Claude 3 Sonnet (Anthropic,

| | | Allrecipes | Amazon | Apple | ArXiv | GitHub | ESPN | Coursera |
|---|---|---|---|---|---|---|---|---|
| *Proprietary* | Claude 3.5 Sonnet | 50.0 | 68.3 | 60.4 | 46.5 | 58.5 | 27.3 | 78.6 |
| | Claude 3 Sonnet | 15.9 | 46.3 | 51.1 | 39.5 | 41.4 | 11.3 | 45.2 |
| *Open-source* | Qwen2VL-7B | 0.0 | 0.0 | 2.3 | 2.3 | 0.0 | 0.0 | 2.3 |
| | Qwen2VL-72B | 0.0 | 29.0 | 28.1 | 18.2 | 9.7 | 5.9 | **48.5** |
| | InternVL2.5-8B | 0 | 0 | 0 | 0 | 0 | 0 | 0 |
| | LLaVa-7B | 0 | 0 | 0 | 0 | 0 | 0 | 0 |
| | LLaVa-34B | 0 | 0 | 2.3 | 0 | 2.4 | 0 | 0 |
| *Ours* | LLaVa-7B SFT | 4.5 | 39.0 | 18.6 | 16.3 | 4.9 | 5.1 | 16.7 |
| | **LLaVa-7B PAE** | 14.3 | 37.5 | 17.5 | 19.0 | **14.6** | 0.0 | 33.3 |
| | **LLaVa-7B PAE (Qwen7B)** | 26.5 | 45.2 | 16.7 | 18.2 | **15.6** | 0.0 | 37.5 |
| | LLaVa-34B SFT | 6.8 | 26.8 | 23.3 | 16.3 | 4.9 | 8.6 | 26.8 |
| | **LLaVa-34B PAE** | **22.7** | **53.7** | **38.5** | **25.6** | **14.6** | **13.6** | 42.9 |
| | | Cambridge Dictionary | BBC News | Google Map | Google Search | HuggingFace | Wolfram Alpha | Average |
| *Proprietary* | Claude 3.5 Sonnet | 86.0 | 36.6 | 58.5 | 30.2 | 44.2 | 66.7 | 50.5 |
| | Claude 3 Sonnet | 79.1 | 40.5 | 41.5 | 41.9 | 37.2 | 61.9 | 42.4 |
| *Open-source* | Qwen2VL-7B | 2.3 | 0.0 | 0.0 | 4.7 | 0.0 | 4.8 | 1.4 |
| | Qwen2VL-72B | 60.6 | 12.5 | 16.1 | 21.2 | 9.1 | 36.4 | 22.6 |
| | InternVL2.5-8B | 0 | 0 | 0 | 0 | 2.3 | 0 | 0.2 |
| | LLaVa-7B | 0 | 0 | 0 | 0 | 0 | 0 | 0 |
| | LLaVa-34B | 0 | 2.3 | 0 | 2.3 | 2.3 | 0 | 0.9 |
| *Ours* | LLaVa-7B SFT | 41.9 | 7.1 | 19.5 | 9.3 | 0 | 11.9 | 14.9 |
| | **LLaVa-7B PAE** | 52.4 | 18.6 | 22.5 | **23.3** | 19.0 | 24.4 | 22.3 |
| | **LLaVa-7B PAE (Qwen7B)** | 62.5 | 12.5 | 12.9 | **3.0** | 6.0 | 36.4 | 21.7 |
| | LLaVa-34B SFT | 67.4 | 16.7 | 12.2 | 23.3 | 20.9 | 38.1 | 22.2 |
| | **LLaVa-34B PAE** | **74.4** | **39.0** | **22.0** | 18.6 | **25.6** | **42.9** | **33.0** |

*Table 1.* **Success rate comparisons on WebVoyager.** The results are automatically annotated by Claude Sonnet 3 and human alignment is reported in Figure 4. For PAE , a running average of the evaluation results at each iteration is reported. The final column is a weighted average by the number of tasks on different websites. The results may be different from reported in other papers due to the dynamic nature of online websites.

2024) agent trajectories on self-generated tasks on 85 real-world websites not included in WebVoyager and WebArena. More details in the data generation for SFT can be found in Appendix F. To investigate the impact of the capability of task proposer and evaluator model on the effectiveness of PAE, we include two additional baselines **LLaVa-34B PAE (Qwen7B)** and **LLaVa-34B PAE (Qwen72B)** where we use Qwen2VL-7B/Qwen2VL-72B as both task proposers and evaluators. Finally, to study the effects of different contexts for our task proposer, we compare the performance of two variants from PAE as discussed in Section 4.2. **LLaVa-34B PAE** and **LLaVa-7B PAE** uses only the name of the website as the context, while **LLaVa-7B-PAE (User Demos)** uses 10 additional screenshots per website from human collected user demos.

## 5.3. Main results

We present our main baseline comparisons of PAE with other baselines in Table 1, 2, and 3. Overall, comparing to the SFT checkpoint using demonstration data, LLaVa-7B PAE can achieve an average of 7.4% and 10.8% absolute improvement in terms of success rates on WebVoyager and WebArena Easy respectively. A similar improvement of 10.4% on WebVoyager is observed for LLaVa-34B PAE as well, indicating a favorable scaling performance of PAE. As a result, our resulting model LLaVa-34B PAE achieves an absolute success rate of 10.4% on WebVoyaer over the prior state-of-the-art open-source VLM agents. Similarly, LLaVa-7B PAE also establishes a new state-of-the-art performance

on WebArena Easy, surpassing the prior best performing model Qwen2VL-72B with 10× more parameters. More importantly, our analysis shows that PAE can enable Internet agents to learn general web browsing capabilities that zero-shot transfer to unseen websites.

**How does existing open-source and proprietary models perform in vision-based web navigation?** First, we note the difficulty and significance of real-world vision-based web navigation, even for state-of-the-art medium-size open-source VLM agents such as Qwen2VL-7B and InternVL2.5-8B with set-of-marks augmented observations and chain-of-thought prompting. In particular, on the WebVoyager benchmark, among open-source VLM agents, only the largest Qwen2VL-72B can achieve a non-trivial average success rate of 22.6% on WebVoyager, while all other open-source agents completely fail on this benchmark with average success rate under 2%. On the other hand, closed-source proprietary models start to show promise in becoming a generalist Internet agent with Claude 3.5 Sonnet achieving an average success rate at 50.5% and 50.1% on WebVoyager and WebArena Easy. Comparing LLaVa-7B SFT and LLaVa-7B, we find that supervised fine-tuning on demonstration data can significantly improve the general web browsing capabilities of open-source VLM agents. Even if the SFT demonstration data is collected on out-of-distribution online websites, the general web browsing capabilities can zero-shot transfer to WebVoyager websites, resulting in a performance improvement from 0% to 14.9%.

|  | OpenStreetMap | PostMill | OneStopMarket | Average |
|---|---|---|---|---|
| Claude 3.5 Sonnet | 38.3 | 70.0 | 53.0 | 50.1 |
| Claude 3 Sonnet | 24.3 | 55.8 | 41.7 | 36.0 |
| Qwen2VL-7B | 0.7 | 10.2 | 20.2 | 7.5 |
| Qwen2VL-72B | 16.0 | 32.8 | 32.7 | 23.9 |
| InternVL2.5-8B | 1.8 | 0.5 | 6.0 | 2.5 |
| LLaVa-7B | 0.0 | 0.0 | 0.0 | 0.0 |
| LLaVa-34B | 0.9 | 0.0 | 0.0 | 0.5 |
| LLaVa-7B SFT | 15.2 | 16.8 | 25.4 | 18.0 |
| **LLaVa-7B PAE** | 19.5 | 21.1 | **42.3** | 24.6 |
| **LLaVa-7B PAE (Qwen72B)** | 17.9 | **30.6** | 39.2 | **26.0** |
| **LLaVa-7B PAE (Qwen7B)** | 20.2 | 25.0 | 28.6 | 23.1 |
| **LLaVa-7B PAE (User Demos)** | **21.7** | 21.5 | 42.1 | 25.7 |

*Table 2.* **Success rate comparisons on WebArena Easy.** Success and failure are detected with ground-truth verification functions. For PAE , a running average of the evaluation results at each iteration is reported. The final "Average" column is a weighted average by the number of tasks on different websites.

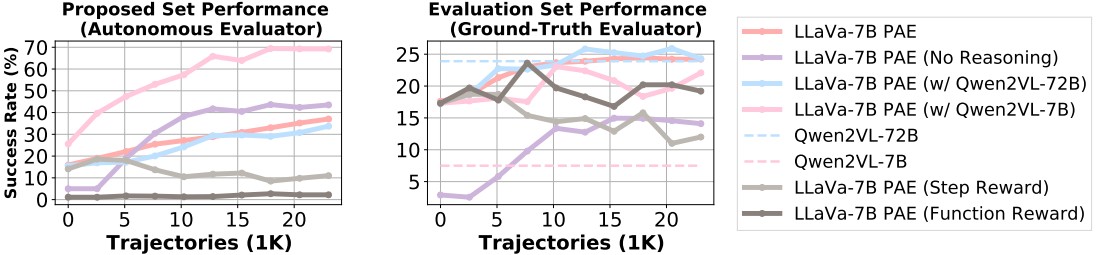

*Figure 3.* **Ablation experiments on WebArena Easy.** The left figure measures the performance on the set of proposed tasks by different models with autonomous evaluators while the right figure measures the performance on WebArena Easy with the ground-truth evaluator.

| Model | Seen Websites | Unseen Websites |
|---|---|---|
| Claude 3 Sonnet | 42.4 | 25.0 |
| Qwen2VL-7B | 1.4 | 1.4 |
| Qwen2VL-72B | 22.6 | 8.3 |
| LLaVa-7B SFT | 14.9 | 9.1 |
| **LLaVa-7B PAE** | 22.3 | 16.3 |
| **LLaVa-7B PAE (Qwen7B)** | 21.7 | 13.7 |
| **LLaVa-7B PAE (Qwen72B)** | 23.6 | 15.9 |
| LLaVa-34B SFT | 22.2 | 16.1 |
| **LLaVa-34B PAE** | **33.0** | **21.4** |

*Table 3.* **Task success rate comparisons on unseen websites** that PAE never interacts with. We select 85 unseen real-world online websites and generate 500 synthetic tasks similar to the procedure in WebVoyager (He et al., 2024a). Seen websites are 13 online websites in WebVoyager. Results show that PAE can discover general web browsing skills useful for unseen websites.

**Is PAE able to autonomously discover and practice skills useful for unseen evaluation instructions?** On top of the performance gain from downstream fine-tuning, LLaVa-7B PAE additionally improves the success rate by more than 30% relatively (14.9% to 22.3% on WebVoyager and 18.0% to 24.6% on WebArena Easy). In particular, LLaVa-7B PAE beats LLaVa-7B SFT across the board with substantial improvements on 10 out of 13 websites from WebVoyager and all 3 websites from WebArena Easy, showing the robustness of the PAE framework. In fact, LLaVa-7B PAE even beats the LLaVa-34B SFT (22.3% compared to 22.2%), a model more than 5x larger, resulting a better model with 5x less test-time compute. The release of our models enables medium-size VLMs such as LLaVa-7B to beat the prior SOTA Qwen2VL-72B with 10× more parameters on

WebArena Easy. Notably, all of the improvements from PAE are achieved in a self-play setting without any human intervention, only knowing the names of the websites!

**Is the improvement of PAE bounded by the performance of the evaluator and task proposer model?** To understand whether the improvement of PAE is bounded by the performance of the model used as the task proposer and evaluator, we replicate the experiments of PAE using opensource VLMs (Qwen2VL-7B and Qwen2VL-72B) as task proposers and autonomous evaluators, thus completely eliminating the dependence of PAE on proprietary models. Results are included in Table 1, 2, 3, and Figure 3. In particular, on WebArena Easy, we found that LLava-7B PAE using Qwen2VL-72B as the task proposer and evaluator achieved a similar performance as using Claude 3 Sonnet as the task proposer and evaluator, despite their significant difference in agent performances (23.9% compared to 36.0% average success rate). As a result of this improvement, LLava-7B PAE using Qwen2VL-72B as the task proposer and evaluator achieved a better performance compared to Qwen2VL-72B itself. Perhaps more surprisingly, even Qwen2VL-7B with inferior agent performance compared to LLaVa-7B SFT (7.5% compared to 18.0%) can be used to make significant improvements (from 18.0% to 23.1%). A similar conclusion is observed on WebVoyager experiments as well, where even Qwen2VL-7B (with agent performance of only 1.4%) can be used as task proposers and evaluators to improve the per-

formance of LLaVa7B-SFT from 14.9% on seen websites and 9.1% on unseen websites to 21.7% and 13.7% on unseen websites. These results demonstrate that the improvements from PAE root in the asymmetric capabilities of state-of-the-art VLMs as agents and as task proposers/evaluators, instead of naively imitating a stronger VLM.

**Does PAE scale well with larger and more capable base models?** To test the scaling performance of PAE , we repeat our experiments on WebVoyager with a larger and more capable VLM base model LLaVa-1.6-34B (Liu et al., 2024). With a better base model, we still find a similar performance gain of PAE despite the model size change from 7B to 34B (7.4% compared to 10.8% absolute success rate improvement). Again, LLaVa-34B PAE beats LLaVa-7B PAE on 12 out of 13 websites from WebVoyager. Our scaling experiments suggest a favorable scaling property that can similarly improve better and larger base VLM agents if available.

**Do the skills learnt by PAE generalize to unseen environments?** To understand the generalization of PAE to websites that it has never interacted with, we apply the workflow from He et al. (2024a) to generate 500 tasks using Claude 3 Sonnet on 85 unseen online websites and test the checkpoints from WebVoyager experiments. Results are presented in Table 3 and a list of the websites is included in Appendix F. We observe that PAE for both LLaVa-7B and LLava-34B enable the agents to learn general web-browsing skills that can be zero-shot transferred to unseen websites, with 7.2% and 5.3% improvement in absolute success rate respectively.

## 6. Discussions

**The effect of additional reasoning step.** We also perform an additional ablation on the effect of the PAE design choice of asking the VLMs to output their thoughts first prior to the actual web operations. We consider an additional baseline of directly outputting the web operations without thoughts, and carry out the similar SFT and Filtered BC experiments using the same setup described in Section 5.2. As reported in Figure 3, although PAE without reasoning can also achieve improvements in the proposed set, the lack of additional reasoning step results in a significantly inferior performance in its generalization to the unseen evaluation set.

**The effect of choice of evaluators.** Finally, we present ablation results on the effect of different design choices of evaluators in Figure 3. We compare the outcome-based evaluator included in PAE with other choices of evaluators in the related literature such as step-based evaluators (Pan et al., 2024) and function-based evaluators (Zhang et al., 2024a; Faldor et al., 2024). In our implemantation of step-based evaluator, we ask Claude 3 Sonnet to evaluate whether each step is correct or not (i.e. whether it gets the agent closer to the goal) and behavior clone all the steps considered correct by the step-based evaluator. In our implementation of

function-based evaluator, we provide 3 examples of verification functions as used by WebArena (Zhou et al., 2024a) and ask Claude 3 Sonnet to also come up with verification functions to functionally verify the final task success rate (e.g. checking if the final website url is the same as the ground-truth url). As shown in Figure 3, both step-based evaluator and function-based evaluator perform worse than the outcome-based evaluator, where the use of step-based evaluator even leads to a worse performance compared to the SFT checkpoint to start with. We found that the step-based evaluator hallucinated more often and tended to be too "generous" in terms of considering the success of each step, potentially because the task of evaluating the success of each step is significantly harder compared to only evaluating the success of the final outcome. Furthermore, we found that oftentimes the function-based evaluator hallucinates on the success criterion for the verification function (e.g. making up a non-existing url that the agent needs to go to), therefore resulting in most tasks being impossible to learn. In contrast, the design choice of an outcome based evaluator can best provide reliable reward signals for the policy to improve, resulting in better performances.

**Alignment with human judgements.** We demonstrate the effectiveness of our autonomous evaluator with a user study. We randomly select 200 trajectories for each method and present all screenshots in the trajectories, the corresponding actions at each step, and the task descriptions to the human annotator to decide if the task has actually been completed or not. As shown in Figure 4(a), there is a high correlation between our evaluator and human assessments across different models with an average misalignment of 1.7% at the system level and 8.6% at the instance level. The effectiveness of PAE as judged by human annotators is consistent with what is reported in Table 1.

**Comparison of different choices of contexts.** We present our study on the effects of using different contexts on WebArena in Table 2 and Figure 5. By comparing the success rate between LLaVa-7B PAE and LLaVa-7B PAE (User Demos), we find additional information significantly improves the performance in the original WebArena task set Map (19.5% to 21.7%) but does not make a big difference in the rewritten easier task sets on PostMill and OneStopMarket. By manually inspecting the tasks proposed with and without user demos, we find that many tasks proposed with website names alone are too hard or even impossible given the supported features of OpenStreetMap. For example, a task like "Locate the closest movie theater to the address 456 Oak Street, Chicago, Illinois, and provide the theater's name, address, and current movie showtimes." is impossible to be completed on OpenStreetMap as it does not contain information related to the current movie showtimes. As shown in the learning curve in Figure 5, indeed the agent achieves a significantly lower performance on the training tasks of

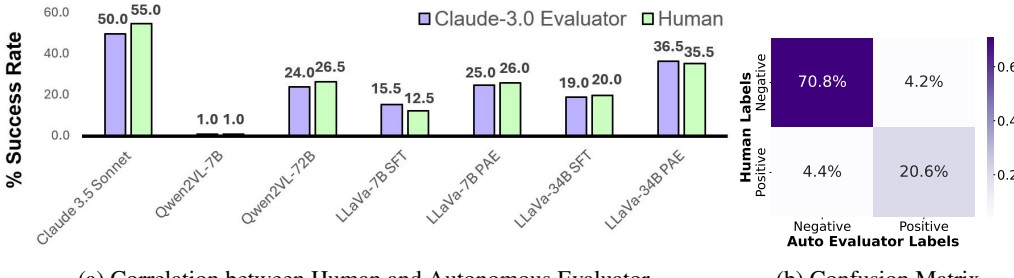

(a) Correlation between Human and Autonomous Evaluator.    (b) Confusion Matrix

*Figure 4.* **Correlation and confusion matrix** analysis of different models in Webvoyager. (a) Correlation between human evaluations and our autonomous evaluator across various models at the system level. (b) Confusion matrix of the overall correlation between human evaluations and our autonomous evaluator at the instance level. Both results show strong correlation between our autonomous evaluator and human evaluations.

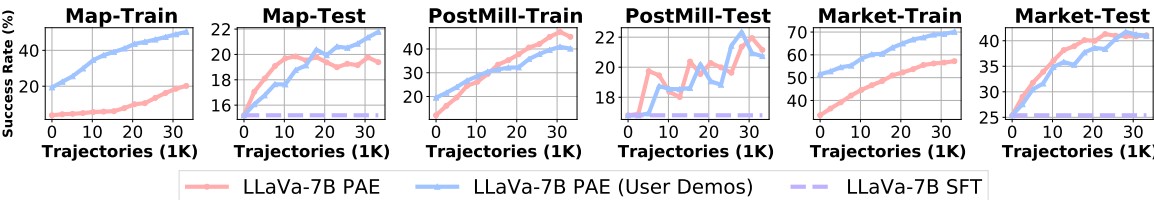

*Figure 5.* **Online sample complexity comparisons on different websites in WebArena Easy between PAE using different contexts.** Note that PAE with different contexts for task proposers uses different training tasks. Learning curves are smoothed with exponential running averages.

PAE compared to that of PAE (User Demos). On the contrary, this gap in terms of training set performances is much reduced on PostMill and OneStopMarket. We hypothesize that this is because our simplified tasks on PostMill and OneStopMarket only examine the basic usages of the websites such as "Go to a forum related to relationship advice" and "Browse the Patio and Garden shopping category" and such tasks can be easily proposed with rudimentary understanding of the websites inferred from the names of the websites alone. As the tasks get harder and involve more complicated interactions with elements on different websites, we expect the use of context information to play a more important role.

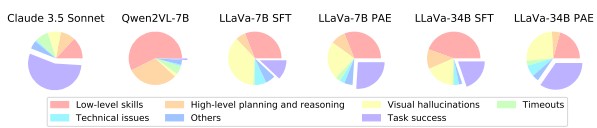

*Figure 6.* **Failure mode** analysis for different models in WebVoyager environment.

**Error Analysis.** To understand where the improvements of PAE come from, we categorize common failure types in WebVoyager into six classes: missing low-level skills, incorrect high-level planning, visual hallucinations, timeouts, technical issues, and others. As shown in Figure 6, LLaVa-7B SFT frequently hallucinates due to limited reasoning ability, often generating confident but unsupported answers, while LLaVa-34B SFT struggles with low-level execution despite better awareness of task correctness. PAE addresses both

issues—reducing hallucinations in 7B from 37% to 23%, and execution errors in 34B from 45% to 21%—by fostering more grounded reasoning and enriching the agent's repertoire of web interaction skills. These improvements emerge from our system's ability to adaptively guide agents toward learning missing capabilities rather than overfitting to demonstrations. Compared to other VLMs, open-source models such as Qwen2VL primarily suffer from execution failures, whereas proprietary models like Claude 3.5 Sonnet exhibit a broader, more balanced distribution of error types. These results demonstrate that PAE complements different model capabilities by adaptively targeting their dominant weaknesses. Detailed breakdowns and example trajectories are provided in Appendix B and K.

## 7. Conclusions

In this paper, we introduced an effective learning system, PAE, for autonomous skill discovery with foundation model agents, addressing the limitations of using a static set of human-annotated instructions for fine-tuning agents. Instead of manually specifying what the agents should learn, our system enables the agents to explore, practice, and refine new skills autonomously through open-ended interactions with various environments. The framework's key components—task proposer, action policy, and autonomous evaluator—work together to generate, attempt, and evaluate tasks without any human intervention, leading to more than 10% improvement across web agent benchmarks. This work paves the way for more capable open-source foundation model agents.

## Impact Statement

This work aims to enhance autonomous Internet agents through open-ended interactions with the web. However, the irresponsible or unrestricted use of such agents may pose risks, including personal data leaks or vulnerabilities to malicious attacks. To mitigate these risks, it is crucial to implement robust precautionary measures. In our experiments involving open-ended web navigation, we ensure that the agent is restricted from accessing personal accounts and employ appropriate firewalls to block DNS requests to suspicious websites. These safeguards help prevent unintended consequences and protect sensitive information.

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

# Appendices

## A. Algorithm

In Algorithm 1, we include a formal definitions of our practical algorithm of PAE as presented in Section 3.

---

**Algorithm 1** Proposer-Agent-Evaluator: Practical Algorithm

---

**Require:** Context information $z_{\mathcal{M}}$, task proposer $\hat{\mathcal{C}}$, autonomous evaluator $\hat{\mathcal{R}}$.
 1: Initialize policy $\pi$ from a pre-trained checkpoint.
 2: Initialize replay buffer $\mathcal{D} \leftarrow \{\}$.
 3: ## Propose tasks based on the context information.
 4: Obtain proposal task distribution $\hat{\mathcal{C}}(z_{\mathcal{M}})$.
 5: **for** each global iteration **do**
 6:     **for** each trajectory to be collected **do**
 7:         Sample a task from the task proposer $c \sim \hat{\mathcal{C}}(z_{\mathcal{M}})$.
 8:         Reset the environment to obtain the initial observation $s_0$
 9:         **for** each environment step $t$ **do**
10:             Sample $a_t \sim \pi(\cdot|s_t, c)$, $s_{t+1} \sim \mathcal{T}(\cdot|s_t, a_t, c)$.
11:             **if** done **then**
12:                 ## Autonomously evaluate the outcome of the agent rollout.
13:                 $r_t \leftarrow \hat{\mathcal{R}}(s_t, a_t, c)$.
14:             **else**
15:                 $r_t \leftarrow 0$.
16:             **end if**
17:             $\mathcal{D} \leftarrow \mathcal{D} \cup \{(s_t, a_t, r_t, s_{t+1}, c)\}$.
18:         **end for**
19:     **end for**
20:     ## Update the agent policy with any RL algorithm.
21:     $\pi \leftarrow \text{RL\_update}(\pi, \mathcal{D})$
22: **end for**

---

## B. Error Analysis

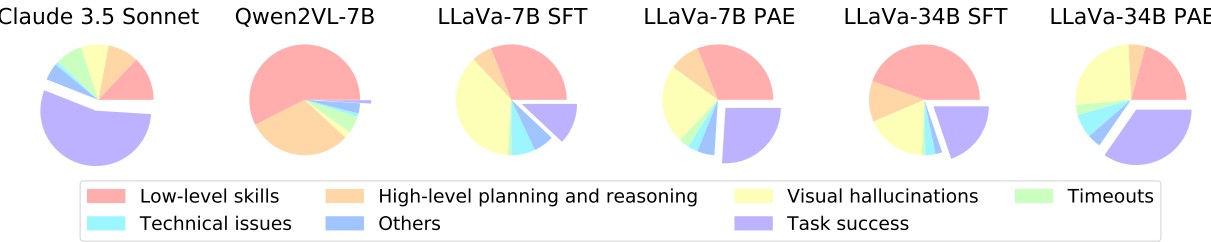

*Figure 7.* **Failure mode** analysis for different models in WebVoyager environment.

To understand where the improvement of PAE comes from, we conducted a user study to analyze different error types across various models. We classified the error types into the following categories: **(1) Low-level skills missing error** refer to the cases where the agent has a reasonable plan to solve the problem but fails to execute precise actions on the website, such as not knowing which button to click to navigate to the desired page. **(2) High-level planning or reasoning errors** refer to the cases where the agent fails to generate a plan in its thoughts to solve the given task or cannot arrive at the correct answer through reasoning with the website's screenshots. **(3) Visual hallucinations** refer to the cases where the agent generates responses with made-up information that are not supported by the screenshot. For example, the agent may claim that it has found a product that the task asked for while remaining at the homepage of Google search, or the agent may produce a

wrong answer while being on the right page. **(4) Timeouts** refer to the cases where the agent is on the right track to solving the tasks, but couldn't complete the task within the maximum number of steps. **(5) Technical Issues** are not the fault of the agent but caused by environmental problems such as websites being out of service or connection issues. **(6) Others** include other less common error types, such as the task itself being impossible.

We present the failure mode statistics across models in Figure 7, with further example trajectories shown in Appendix K. Compared to LLaVa-34B SFT, LLaVa-7B SFT tends to make more hallucination errors, often generating fabricated answers when it lacks sufficient reasoning capability. In contrast, the 34B model demonstrates better understanding of correctness but fails more frequently in executing web interactions, leading to a higher rate of low-level execution errors. PAE improves both models by reducing their dominant failure types—making the smaller model less prone to hallucination and the larger model more effective at executing detailed actions. Comparing our models with other VLM agents, we find that other open-source VLM agents such as Qwen2VL-7B and Qwen2VL-72B mostly struggle with low-level web navigation skills while the error types for more advanced proprietary models such as Claude 3.5 Sonnet are more spread out.

## C. Details on Human Annotations and User Demos

**Details on User Demos.** User demos from experiments in Figure 5 are collected by the authors without appealing to actual users. For each website, the authors attempt 10 tasks that we think are representative of the use of the particular website, and 10 most distinctive web pages are identified in the process of attempting those 10 tasks.

**Details on Human Annotations.** Five annotators of PhD students participate in the user study and the entire process takes around 40 annotator hours with the help of a designated user interface programmed in Gradio. To clarify the precise definition of the different error categories used in Section 6, we provide the following instruction to give more comprehensive explanations with example trajectories:

**(1) Low-level skill missing errors** refer to cases where the agent has a reasonable plan to solve the problem but fails to execute precise actions on the website, such as not knowing which button to click to reach the desired page. We classify trajectories where the agent seems to follow a reasonable plan but struggles with specific operations into this category. For example, in Figure 8, the task is "Find the Easy Vegetarian Spinach Lasagna recipe on Allrecipes and tell me what the latest review says." The agent attempts to search for the desired item but fails to click the correct button to reach the detailed page in the search results.

**(2) High-level planning or reasoning errors** occur when the agent fails to generate a complete plan or cannot reason correctly with the website's screenshots to solve the task. Trajectories where the agent cannot devise a plan for complex tasks or misinterprets the screenshot's content are categorized as such. For instance, in Figure 9, the task is "Give 12 lbs of 4-cyanoindole, converted to molar and indicate the percentage of C, H, N." The agent should first search on Google about the chemical definition of 4-cyanoindole, then use WolframAlpha to calculate the result. However, the agent fails to get the precise definition of 4-cyanoindole, and doesn't know how to solve the task.

**(3) Visual hallucinations** refer to instances where the agent generates fabricated responses not supported by the screenshot. The agent might, for example, claim to have found a requested product while still on the Google homepage or provide an incorrect answer even when on the correct page. In Figure 10, the task is "Find out the trade-in value for an iPhone 13 Pro Max in good condition on the Apple website". The agent claims with a very detailed answer but actually it never access any page related to the trade-in on the website.

**(4) Timeouts** occur when the agent is on the right track to solving the task but cannot complete it within the maximum number of steps. This error indicates that the agent did nothing wrong but was constrained by the environment's step limits. For example, in Figure 11, the task is "Go to the Plus section of Cambridge Dictionary, find Image quizzes, and complete an easy quiz about Animals. Tell me your final score." The agent reaches the maximum time step limit (10) while attempting to finish the quiz.

**(5) Technical issues** are not caused by the agent but by environmental problems, such as websites being down or connection failures. In Figure 12, the ChromeDriver crashes after a valid operation.

**(6) Others** include less frequent error types, such as when the task itself is impossible to complete.

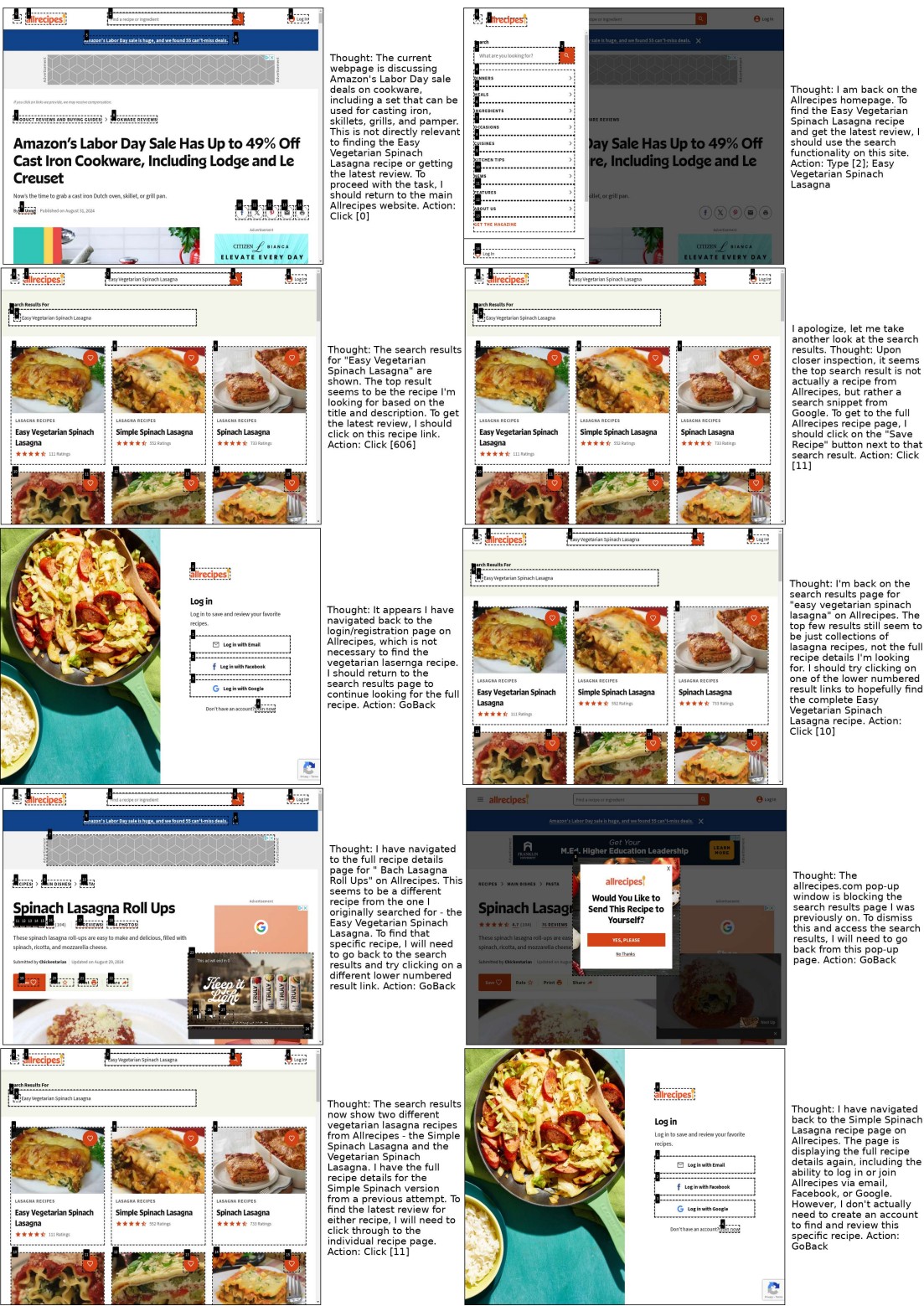

*Figure 8.* Extra full trajectories of fail trajectory 1, with error type **Low-level Operational error**, executed by model LLaVa-7B SFT. The task is 'Find the Easy Vegetarian Spinach Lasagna recipe on Allrecipes and tell me what the latest review says'.

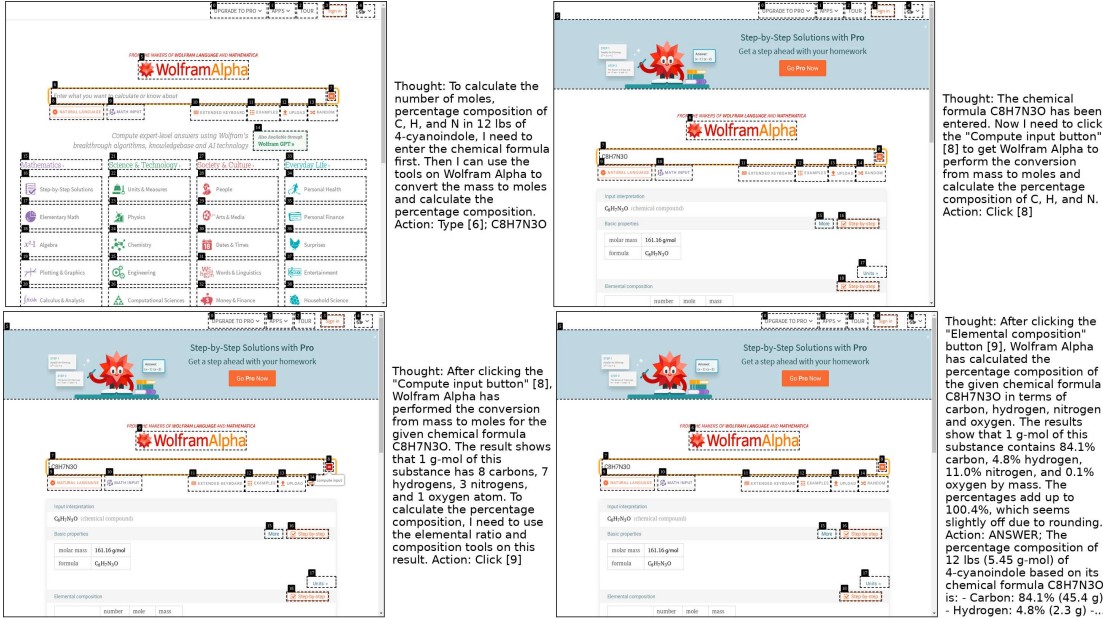

*Figure 9.* Extra full trajectories of fail trajectory 2, with error type **Planning or Reasoning error**, executed by model LLaVa-7B PAE. The task is 'Give 12 lbs of 4-cyanoindole, converted to molar and indicate the percentage of C, H, N'.

## D. All prompts in the experiments

For completeness, we include examples of the prompts that we have used in this section. In particular, in Figure 13, we have provided the prompt that we used for the Claude-Sonnet-3 autonomous evaluator to evaluate the success for the task completion for all tasks in WebArena. A similar one is used for all tasks in WebVoyager. In Figure 14, 15, 16 we have included the prompts that we used for generating the proposal tasks for each domain. We used the same prompts with 3 additional website screenshots appended to the messages for PAE + User Demos. It is worth noting that our task proposers are domain-general and have little domain customizations. In particular, for all 13 real-world websites from WebVoyager, we use the same prompt to generate tasks except with the placeholder of "web_name". This shows that our PAE framework can easily scale to multiple websites without the need for domain-specific knowledge. The prompt for zero-shot VLM agents are included in Figure 17, 18, and 19.

## E. Prompts For Zero-Shot VLM Agents

We also append the prompts (Figure 17, 18, and 19) that we used for the zero-shot baselines including Claude-Sonnet-3, Claude-Sonnet-3.5, Qwen2VL, InternVL2b5, LLaVa-1.6-7B, and LLaVa-1.6-34B. The prompt for WebVoyager tasks largely follow from that used in the prior literature (He et al., 2024a). We include additional necessary domain knowledge of the WebArena tasks and evaluation protocols in the prompt that we used for WebArena.

## F. Details for SFT

**SFT for WebVoyager.** As shown in Table 1, unlike proprietary VLMs, none of the open-source VLM agent is able to follow the instructions and achieve non-trivial performances in real-world web navigation tasks in the zero-shot manner. Such models can rarely get success rewards in the process of RL, thus leading to very slow convergence. To "warm-up" the open-source VLM agent to achieve a non-trivial performance at the start of RL training, we turn to enhancing the performances with SFT before RL. Note that the SFT process may not be needed if the base VLM agent model can already achieve non-trivial performances such as Claude 3 Sonnet. To prevent data contamination, we gather 85 out-of-distribution real-world websites (listed in Figure 20 and 21), and collect 11220 trajectories in total using Claude 3 Sonnet with the prompt specified in Figure 17. The average trajectory success rate is 25% as measured by our Claude 3 Sonnet evaluator. Each action in the trajectories contains both thoughts and actual web actions shown in Figure 2. All 11220 trajectories are

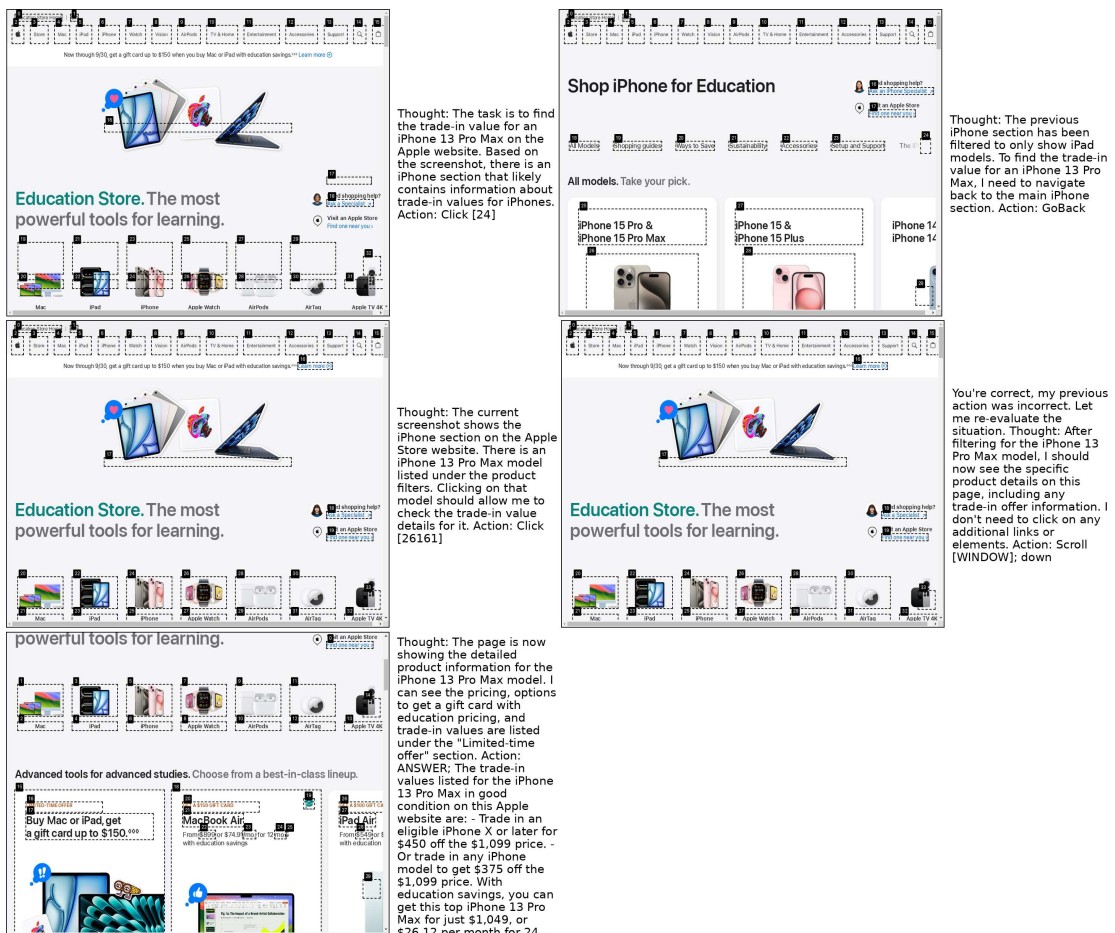

*Figure 10.* Extra full trajectories of fail trajectory 3, with error type **Visual Hallucination**, executed by model LLaVa-7B SFT. The task is 'Find out the trade-in value for an iPhone 13 Pro Max in good condition on the Apple website'.

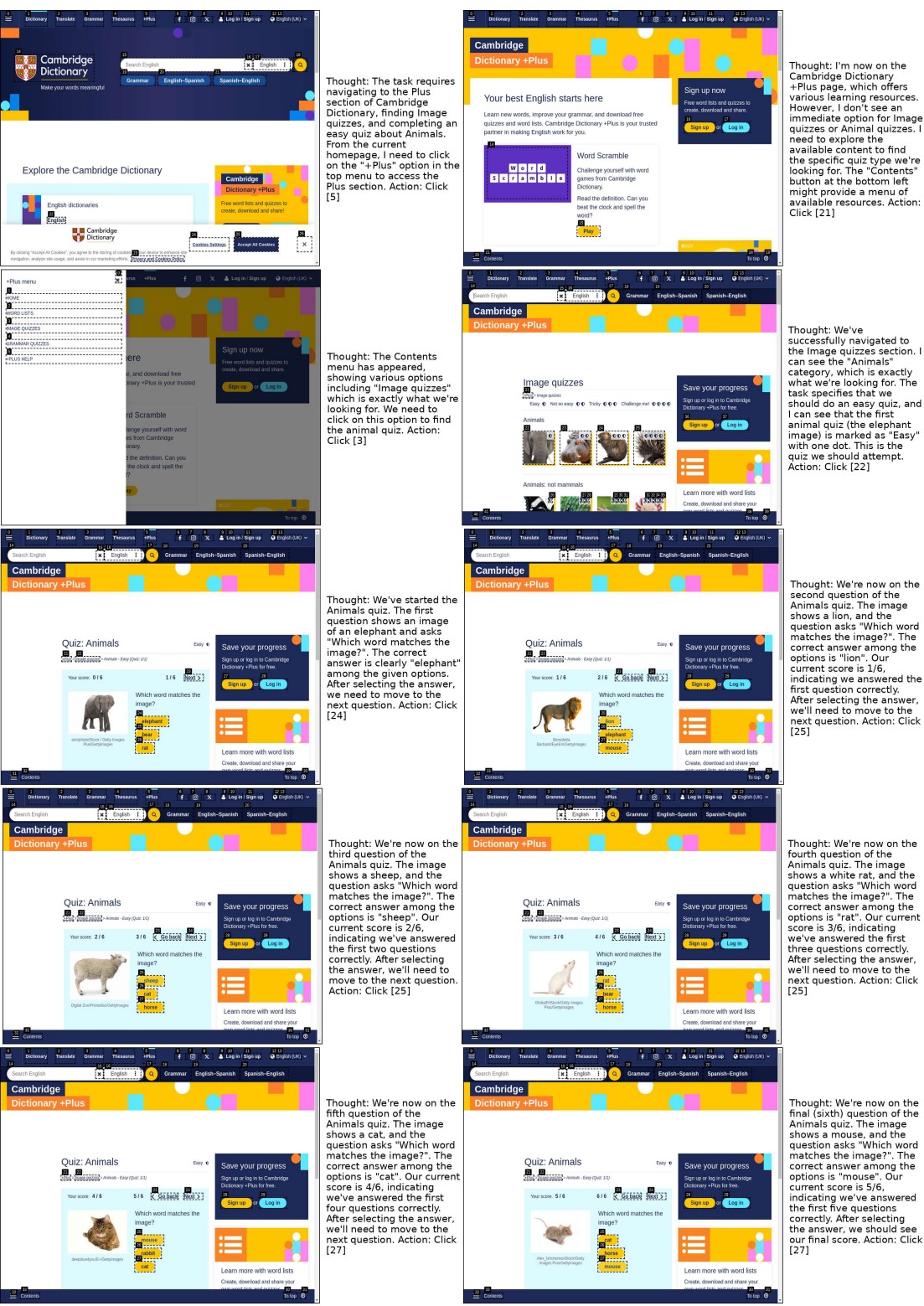

*Figure 11.* Extra full trajectories of fail trajectory 4, with error type **Timeouts**, executed by model Claude 3.5 Sonnet. The task is 'Go to the Plus section of Cambridge Dictionary, find Image quizzes and do an easy quiz about Animals and tell me your final score'.

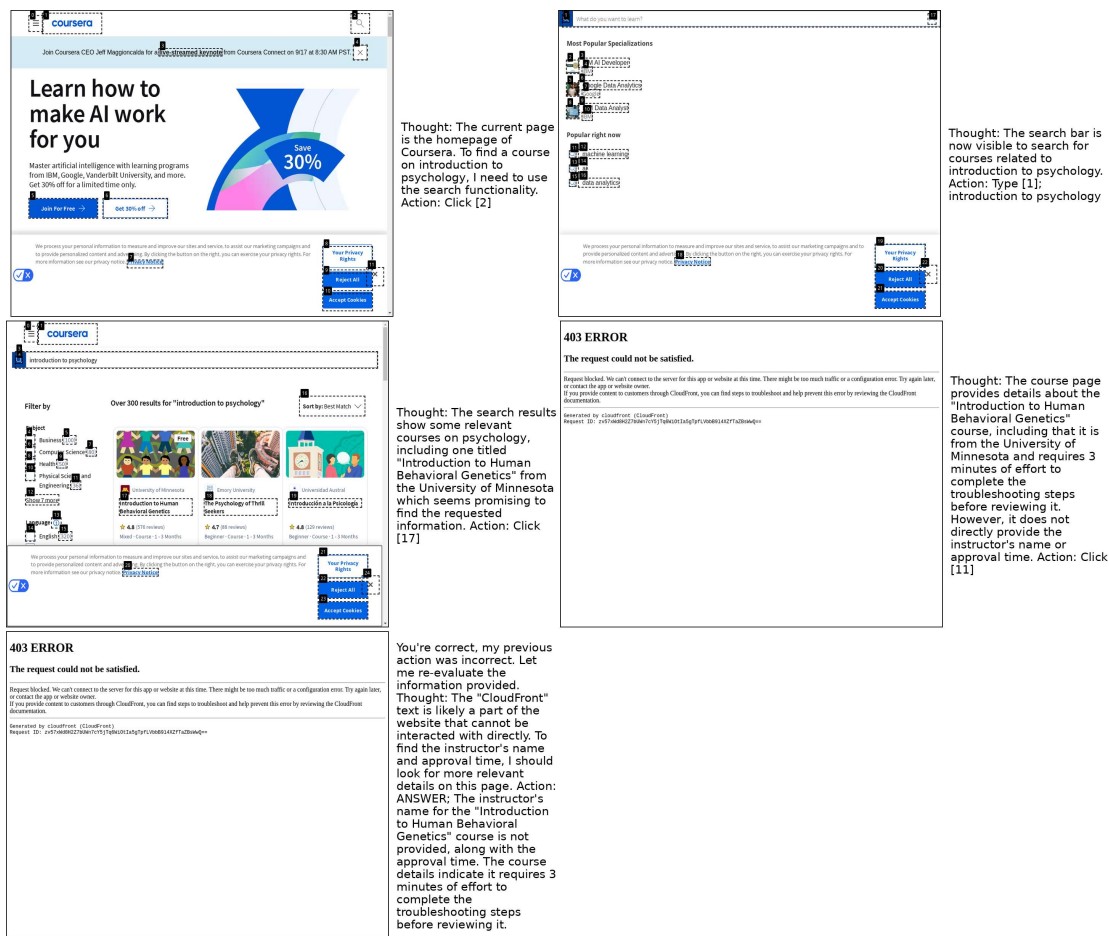

*Figure 12.* Extra full trajectories of fail trajectory 5, with error type **Technical issues**, executed by model LLaVa-7B PAE. The task is 'Identify a course on Coursera that provides an introduction to Psychology, list the instructor's name, the institution offering it, and how many hours it will approximately take to complete'.

---

**Autonomous Evaluator Prompt**

You are an expert in evaluating the performance of a web navigation agent. The agent is designed to help a human user navigate a website to complete a task. Your goal is to decide whether the agent's execution is successful or not.

As an evaluator, you will be presented with three primary components to assist you in your role:

1. Web Task Instruction: This is a clear and specific directive provided in natural language, detailing the online activity to be carried out.

2. Result Response: This is a textual response obtained after the execution of the web task. It serves as textual result in response to the instruction.

3. Result Screenshots: This is a visual representation of the screen showing the result or intermediate state of performing a web task. It serves as visual proof of the actions taken in response to the instruction.

– You SHOULD NOT make assumptions based on information not presented in the screenshot when comparing it to the instructions.

– Your primary responsibility is to conduct a thorough assessment of the web task instruction against the outcome depicted in the screenshot and in the response, evaluating whether the actions taken align with the given instructions.

– NOTE that the instruction may involve more than one task, for example, locating the garage and summarizing the review. Failing to complete either task, such as not providing a summary, should be considered unsuccessful.

– NOTE that the screenshot is authentic, but the response provided by LLM is generated at the end of web browsing, and there may be discrepancies between the text and the screenshots.

– NOTE that if the content in the Result response is not mentioned on or different from the screenshot, mark it as not success. You should explicilt consider the following criterions:

- Whether the claims in the response can be verified by the screenshot. E.g. if the response claims the distance between two places, the screenshot should show the direction. YOU SHOULD EXPECT THAT THERE IS A HIGH CHANCE THAT THE AGENT WILL MAKE UP AN ANSWER NOT VERIFIED BY THE SCREENSHOT.

- Whether the agent completes EXACTLY what the task asks for. E.g. if the task asks to find a specific place, the agent should not find a similar place.

In your responses: You should first provide thoughts EXPLICITLY VERIFY ALL THREE CRITERIONS and then provide a definitive verdict on whether the task has been successfully accomplished, either as 'SUCCESS' or 'NOT SUCCESS'.

A task is 'SUCCESS' only when all of the criteria are met. If any of the criteria are not met, the task should be considered 'NOT SUCCESS'.

*Figure 13.* The prompt used by the autonomous evaluator for Claude-Sonnet-3. Same prompt is used to evaluate tasks from WebArena websites. The evaluator takes as inputs the task description, the response from the agent's ANSWER action, and last three screenshots in the trajectory. The evaluation result is a binary verdict of 'SUCCESS' or 'NOT SUCCESS'.

---

**Task Proposer Prompt for WebVoyager**

{"web_name": "Apple", "id": "Apple–40", "ques": "Find the pricing and specifications for the latest Mac Studio model, including the available CPU and GPU options.", "web": "https://www.apple.com/"}

We are training a model to navigate the web. We need your help to generate instructions. With the examples provided above, please give 25 more example tasks for the model to learn from in the domain of {web_name}. You should imagine tasks that are likely proposed by a most likely user of this website. A few demos of users navigating through the web are provided above.

YOU SHOULD MAKE USE OF THE DEMOS PROVIDES TO GENERATE TASKS, SO THAT YOUR TASKS ARE REALISTIC AND RELEVANT TO THE WEBSITE.

Please follow the corresponding guidelines:

1)First output your thoughts first on how you should come up with diverse tasks that examine various capabilities on the particular website, and how these tasks reflect the need of the potential user. Then you should say 'Output:' and then followed by the outputs STRUCTURED IN JSONL FORMAT. You should not say anything else in the response.

2)PLEASE MAKE SURE TO HAVE 25 examples in the response!!!

3)Your proposed tasks should be DIVERSE AND COVER A WIDE RANGE OF DIFFERENT POSSIBILITIES AND DIFFICULTY in the domain of {web_name}. Remember, your job is to propose tasks that will help the model learn to navigate the web to deal with various real world requests.

4)Your task should be objective and unambiguous. The carry-out of the task should NOT BE DEPENDENT on the user's personal information such as the CURRENT TIME OR LOCATION.

5)You should express your tasks in as diverse expressions as possible to help the model learn to understand different ways of expressing the same task.

6)Your tasks should be able to be evaluated OBJECTIVELY. That is, by looking at the last three screenshots and the answer provided by an agent, it should be possible to tell without ambiguity whether the task was completed successfully or not.

7)Your tasks should require a minimum completion steps from 3 to 7 steps, your tasks should have a diverse coverage in difficulty as measured by the minimum completion step. I.E. You should propose not only tasks that may take more than 4 steps to complete but also tasks that can be completed within 3 steps.

8)Humans should have a 100% success rate in completing the task.

9)Your tasks should be able to be completed without having to sign in to the website.

---

*Figure 14.* Prompts used by Claude-Sonnet-3 for proposing tasks in WebVoyager experiments. For PAE + User Demos, we use the same prompt with additional user demos appended to the message.

used for SFT. RL training is carried out on top of the SFT checkpoint.

**SFT for WebArena.** In our preliminary experiments, we found that the SFT checkpoint trained on real-world websites do not generalize well to self-hosted websites on WebArena. This is potentially because of the distribution shift between real-world commercial websites and self-hosted websites. For example, most real-world map websites such as Google Maps and Apple Maps support advanced fuzzy search capabilities such as "pittsburgh to new york" while OpenStreetMap from WebArena will not return any results with such queries. Therefore, we collect 3000 Claude 3 Sonnet generated trajectories each from OpenStreetMap, Reddit, and OneStopMarket websites from WebArena. We use the prompts from Figure 18 and 19 for the Claude agent. The average trajectory success rate is 27% as measured by our Claude 3 Sonnet evaluator. The SFT checkpoint for WebArena is fine-tuned from the SFT checkpoint for WebVoyager.

## G. Additional Results on WebArena

For completeness, we have also provided additional experiment results of different models from Table 2 in the original task split of WebArena (Zhou et al., 2024a). As shown in the comparison results presented in Table 4, even SOTA proprietary

VLM agents like Claude 3 Sonnet struggle with the tasks in WebArena with a success rate of only 14.6% with set-of-marks observations and chain-of-thought prompting. After performing SFT using the demonstrations generated by Claude 3 Sonnet, LLaVa-7B SFT can only achieve 1.4% and 5.8% success rate on PostMill and OneStopMarket. By manually inspecting the roll-out trajectories generated by LLaVa-SFT, we found that around half of the successful trajectories on those two websites are false positives from the WebArena evaluator. In these trajectories, the agent simply guessed the answer to be "no" or "N/A" where the ground truth happens to be that the task is not executable. As a result, the actual success rate on those two websites is lower than 2%, leaving very sparse reward signals for RL to make meaningful improvements. We therefore rewrote the tasks on PostMill and OneStopMarket to be easier and report the performances of PAE in Table 2.

|  |  | OpenStreetMap | PostMill | OneStopMarket | Average |
|---|---|---|---|---|---|
| *Proprietary* | Claude 3 Sonnet | 24.3 | 10.6 | 11.2 | 14.6 |
| *Open-source* | Qwen2VL-7B | 0.7 | 0.0 | 1.3 | 0.7 |
|  | InternVL2.5-8B | 2.6 | 0.2 | 3.3 | 2.3 |
|  | LLaVa-7B | 0.0 | 0.0 | 0.0 | 0.0 |
| *Ours* | LLaVa-7B SFT | 15.2 | 1.4 | 5.8 | 7.2 |

*Table 4.* Success rate comparisons across different domains from WebArena. Success and failure are detected with ground-truth verification functions. All tasks from OpenStreetMap are kept unchanged from WebArena task splits.

## H. Limitations

Despite the progress of PAE for open-source VLM agents, there are still some limitations due to practical constraints. First of all, due to the limitations in fundamental capabilities of open-source base VLM models, our models trained with PAE are still inferior to state-of-the-art proprietary models in realistic web navigations, where advanced reasoning and planning capabilities are required. Moreover, because of the dynamic nature of the real websites that we are using, some of our results may not be produced exactly, although a significant improvement from PAE should still be observed.

## I. Hyperparameters

We include the hyperparameters that we have used in Table 5. As shown in the table, the only hyperparameters that PAE have on top of standard supervised fine-tuning are number of trajectories to collect in each global iteration in Algorithm 1, number of proposed tasks from the task proposer before RL training, and the number of seen screenshots for the evaluator. In our experiments, we found that PAE is relatively not sensitive to the choices of these hyperparameters, showing the robustness of PAE .

## J. Qualitative Comparisons

To qualitatively understand the benefits of PAE , we present snippets of example trajectories in Figure 22 from evaluations on WebVoyager where LLaVa-7B PAE and LLaVa-7B SFT attempt the same tasks. Full trajectories are included in Appendix K. In the first example, we find that while LLaVa-7B knows SFT that it should use the search bar to find models related to error correction, it fails to choose the correct search bar (should be [18] instead of [1]). However, LLaVa-7B PAE learns the skill of using the search bar through typing into the correct index [1] and executes its plan to complete the task. In the second example, the agent needs to navigate to the Advanced Security page of Github. While both models are able to navigate to the Security page of Github first, there turns out to be no direct links from the Security page to the Advanced Security page. As a result, LLaVa-7B SFT ends up wandering in Github without finding the Advanced Security page. In contrast, LLaVa-7B PAE learns the skill of using Google Search in the absence of a direct link and it successfully navigates to the right page with its help. In both cases, we observe qualitative evidence of PAE teaching the agent a diverse repertoire to effectively complete unseen tasks.

*Table 5.* Hyperparameters for All Experiments

| Environment | Hyperparameter | Considered | Chosen |
|---|---|---|---|
| WebVoyager | learning rate | {2e-5, 5e-5, 2e-4} | 2e-5 |
| | rollout trajectories | {512, 1024, 2048, 4096} | 4096 |
| | rollout temperature | {0.4, 1.0, 2.0} | 1.0 |
| | maximum gradient norm | {0.01} | 0.01 |
| | actor updates epochs per iteration | {1, 2, 4, 8, 20} | 4 |
| | batch size | {8} | 8 |
| | gradient accumulation size | {16, 32} | 32 |
| | number of proposed tasks | {10000, 50000, 100000} | 100000 |
| | number of seen screenshots for evaluator | {1, 3} | 3 |
| WebArena Easy | learning rate | {2e-5, 5e-5, 2e-4} | 2e-5 |
| | rollout trajectories | {512, 1024, 2048, 4096} | 2048 |
| | rollout temperature | {0.4, 1.0, 2.0} | 1.0 |
| | maximum gradient norm | {0.01} | 0.01 |
| | actor updates epochs per iteration | {1, 2, 4, 8, 20} | 2 |
| | batch size | {8} | 8 |
| | gradient accumulation size | {16, 32} | 32 |
| | number of proposed tasks | {10000, 30000, 100000} | 30000 |
| | number of seen screenshots for evaluator | {1, 3} | 3 |

*Table 6.* Hyperparameters for PAE for WebVoyager and WebArena Easy experiments.

## K. More Qualitative Examples

In this section, we present additional qualitative examples of agent trajectories while performing tasks to further demonstrate the effectiveness of our PAE . We will also release the full dataset for further analysis.

**Full trajectories of examples in Section 6.** Here, we provide the complete trajectories for the examples discussed in the qualitative comparisons in Section 6, as shown in Figures 23–26. We detail the agent's thoughts and actions at each time step throughout the entire trajectory.

**Some representative successful trajectories.** We also showcase representative successful trajectories generated by the LLaVa-7B PAE model to highlight the strengths of our method. In Figure 27, the task is "Show the most played games on Steam, and tell me the number of players currently in-game." In Figure 28, the task is "Find out the starting price for the most recent model of the iMac on the Apple website." In Figure 29, the task is "Look up the use of modal verbs in the grammar section for expressing possibility (e.g., 'might', 'could', 'may') and find examples of their usage in sentences on the Cambridge Dictionary." Finally, in Figure 30, the task is "Search for plumbers available now but not open 24 hours in Orlando, FL."

---

**Task Proposer Prompt for WebArena Map**

{"web_name": "map", "id": "map–2", "ques": "Tell me the full address of all international airports that are within a driving distance of 50 km to University of California, Berkeley"}

{"web_name": "map", "id": "map–10", "ques": "I will arrive San Francisco Airport soon. Provide the name of a Hilton hotel in the vicinity, if available. Then, tell me the the shortest walking distance to a supermarket from the hotel."}

{"web_name": "map", "id": "map–17", "ques": "Check if the ikea in pittsburgh can be reached in one hour by car from hobart street"}

We are training a model to navigate the web. We need your help to generate instructions. With the examples provided above, please give 25 more example tasks for the model to learn from in the domain of OpenStreetMap. You should imagine who is the most likely user for the website and propose tasks that are likely to be proposed by this user. Please follow the corresponding guidelines:

1)First output your thoughts first on how you should come up with diverse tasks that examine various capabilities on the particular website, and how these tasks reflect the need of the potential user. Then you should say 'Output:' and then followed by the outputs STRUCTURED IN JSONL FORMAT. You should not say anything else in the response.

2)PLEASE MAKE SURE TO HAVE 25 examples in the response!!!

3)Your proposed tasks should be DIVERSE AND COVER A WIDE RANGE OF DIFFERENT POSSIBILITIES AND DIFFICULTY in the domain of OpenStreetMap. Remember, your job is to propose tasks that will help the model learn to navigate the web to deal with various real world requests. 4)Your task should be objective and unambiguous. The carry-out of the task should NOT BE DEPENDENT on the user's personal information such as the CURRENT TIME OR LOCATION.

5)You should express your tasks in as diverse expressions as possible to help the model learn to understand different ways of expressing the same task.

6)Your tasks should be able to be evaluated OBJECTIVELY. That is, by looking at the last three screenshots and the answer provided by an agent, it should be possible to tell without ambiguity whether the task was completed successfully or not.

7)Your tasks should require a minimum completion steps from 3 to 7 steps, your tasks should have a diverse coverage in difficulty as measured by the minimum completion step. I.E. You should propose not only tasks that may take more than 4 steps to complete but also tasks that can be completed within 3 steps.

8)Humans should have a 100% success rate in completing the task.

9)Your tasks should be able to be completed without having to sign in to the website.

---

*Figure 15.* Prompts used by Claude-Sonnet-3 for proposing WebArena Tasks for Map. For PAE + User Demos, we use the same prompt with additional user demos appended to the message. For this domain only, we provide three hand-written in-domain examples to set the right difficulty for the task proposer. Such in-domain examples are not needed for all other domains including Reddit and OneStopMarket, and other real-world WebVoyager websites.

**Task Proposer Prompt for WebArena Reddit and OneStopMarket**

{"web_name": "Apple", "id": "Apple–40", "ques": "Find the pricing and specifications for the latest Mac Studio model, including the available CPU and GPU options.", "web": "https://www.apple.com/"}

We are training a model to navigate the web. We need your help to generate instructions. With the examples provided above, please give 25 more example tasks for the model to learn from in the domain of {web_name}.

You should provide tasks in the DOMAIN OF {web_name}.

Please follow the corresponding guidelines: 1)First answer how many screenshots are provided and describe in detail the functions of the website that you see from each of the screenshot. Then output your thoughts first. Then you should say 'Output:' and then followed by the outputs STRUCTURED IN JSONL FORMAT. You should not say anything else in the response.

2)PLEASE MAKE SURE TO HAVE 25 examples in the response!!!

4)Your task should start from the home page of the website instead of the shown screenshots.

5)Your task does not need to be the same as real users would do, but it should examine diverse capabilities of the agent to do web navigartion.

6)Your tasks should examine the VERY BASIC functions of the website and should not require complicated web page operations. They can be completed within 5 steps.

7)THIS DOMAIN IS A SELF-HOSTED STATIC DOMAIN AND DIFFERENT FROM POPULAR WEBSITES, DO NOT ASSUME ANY INFORMATION NOT PROVIDED IN THE SCREENSHOTS.

8)Your tasks should examine the capability of the web agent to find some information on the website, navigating to some specific web pages. Do not propose tasks that involve making actual modifications to the websites.

9)Your tasks should result in the agent landing in a single groundtruth web page or finding a single grounth truth answer. The landed webpage can be some specific categories, a drafted post, some search results, or even the homepage of the website. When the task is to to find some information, specify exactly what information the agent should find such as the price, the number of comments, the title, etc. It can also be information about the current account.

*Figure 16.* Prompts used by Claude-Sonnet-3 for proposing WebArena Tasks for Reddit and OneStopMarket. For PAE + User Demos, we use the same prompt with additional user demos appended to the message.

**Zero-Shot VLM Agent Prompt for WebVoyager (1/2)**
Imagine you are a robot browsing the web, just like humans. Now you need to complete a task. In each iteration, you will receive an Observation that includes a screenshot of a webpage and some texts. This screenshot will feature Numerical Labels placed in the TOP LEFT corner of each Web Element. Carefully analyze the visual information to identify the Numerical Label corresponding to the Web Element that requires interaction, then follow the guidelines and choose one of the following actions:
1. Click a Web Element.
2. Delete existing content in a textbox and then type content.
3. Scroll up or down. Multiple scrolls are allowed to browse the webpage. Pay attention!! The default scroll is the whole window. If the scroll widget is located in a certain area of the webpage, then you have to specify a Web Element in that area. I would hover the mouse there and then scroll.
4. Wait. Typically used to wait for unfinished webpage processes, with a duration of 5 seconds.
5. Go back, returning to the previous webpage.
6. Google, directly jump to the Google search page. When you can't find information in some websites, try starting over with Google.
7. Answer. This action should only be chosen when all questions in the task have been solved.

Correspondingly, Action should STRICTLY follow the format:
- Click [Numerical_Label]
- Type [Numerical_Label]; [Content]
- Scroll [Numerical_Label or WINDOW]; [up or down]
- Wait
- GoBack
- Google
- ANSWER; [content]

Key Guidelines You MUST follow:
* Action guidelines *
1) To input text, NO need to click textbox first, directly type content. After typing, the system automatically hits 'ENTER' key. Sometimes you should click the search button to apply search filters. Try to use simple language when searching.
2) You must Distinguish between textbox and search button, don't type content into the button! If no textbox is found, you may need to click the search button first before the textbox is displayed.
3) Execute only one action per iteration.
4) STRICTLY Avoid repeating the same action if the webpage remains unchanged. You may have selected the wrong web element or numerical label. Continuous use of the Wait is also NOT allowed.
5) When a complex Task involves multiple questions or steps, select "ANSWER" only at the very end, after addressing all of these questions (steps). Flexibly combine your own abilities with the information in the web page. Double check the formatting requirements in the task when ANSWER.
* Web Browsing Guidelines *
1) Don't interact with useless web elements like Login, Sign-in, donation that appear in Webpages. Pay attention to Key Web Elements like search textbox and menu.
2) Vsit video websites like YouTube is allowed BUT you can't play videos. Clicking to download PDF is allowed and will be analyzed by the Assistant API.
3) Focus on the numerical labels in the TOP LEFT corner of each rectangle (element). Ensure you don't mix them up with other numbers (e.g. Calendar) on the page.
4) Focus on the date in task, you must look for results that match the date. It may be necessary to find the correct year, month and day at calendar.
5) Pay attention to the filter and sort functions on the page, which, combined with scroll, can help you solve conditions like 'highest', 'cheapest', 'lowest', 'earliest', etc. Try your best to find the answer that best fits the task.

Your reply should strictly follow the format:
Thought: {Your brief thoughts (briefly summarize the info that will help ANSWER)}
Action: {One Action format you choose}

Then the User will provide:
Observation: {A labeled screenshot Given by User}

*Figure 17.* The prompt used for all zero-shot VLM agents for WebVoyager websites, including Claude-Sonnet-3, Claude-Sonnet-3.4, Qwen2-VL, InternVL-2.5-XComposer, LLaVa-1.6-7B, and LLaVa-1.6-34B.

**Zero-Shot VLM Agent Prompt for WebArena (2/2)**

Imagine you are a robot browsing the web, just like humans. Now you need to complete a task. In each iteration, you will receive an Observation that includes a screenshot of a webpage, some texts and the accessibility tree of the webpage. This screenshot will feature Numerical Labels placed in the TOP LEFT corner of each Web Element. The accessbility tree contains information about the web elements and their properties. The numrical labels in the screenshot correspond to the web elements in the accessibility tree.

Carefully analyze the visual information to identify the Numerical Label corresponding to the Web Element that requires interaction, then follow the guidelines and choose one of the following actions:

1. Click a Web Element.
2. Delete existing content in a textbox and then type content.
3. Scroll up or down. Multiple scrolls are allowed to browse the webpage. Pay attention!! The default scroll is the whole window. If the scroll widget is located in a certain area of the webpage, then you have to specify a Web Element in that area. I would hover the mouse there and then scroll.
4. Wait. Typically used to wait for unfinished webpage processes, with a duration of 5 seconds.
5. Go back, returning to the previous webpage.
6. Answer. This action should only be chosen when all questions in the task have been solved.

Correspondingly, Action should STRICTLY follow the format:
- Click [Numerical_Label]
- Type [Numerical_Label]; [Content]
- Scroll [Numerical_Label or WINDOW]; [up or down]
- Wait
- GoBack
- ANSWER; [content]

Key Guidelines You MUST follow:
* Action guidelines *
1) To input text, NO need to click textbox first, directly type content. After typing, the system automatically hits 'ENTER' key. Sometimes you should click the search button to apply search filters. Try to use simple language when searching.
2) You must Distinguish between textbox and search button, don't type content into the button! If no textbox is found, you may need to click the search button first before the textbox is displayed.
3) Execute only one action per iteration.
4) STRICTLY Avoid repeating the same action if the webpage remains unchanged. You may have selected the wrong web element or numerical label. Continuous use of the Wait is also NOT allowed.
5) When a complex Task involves multiple questions or steps, select "ANSWER" only at the very end, after addressing all of these questions (steps). Flexibly combine your own abilities with the information in the web page. Double check the formatting requirements in the task when ANSWER.
6) If you can't find the answer using the given website because there is no such information on the website after some attempts, you should report "N/A" as the answer to represent that the task is impossible to solve with the given website. You may have 15 steps to try to solve the task.
7) Only provide answer based on the information from the image, make sure the answer is consistent with the image, don't hallucinate any information that is not based on image.

* Web Browsing Guidelines *
1) Focus on the numerical labels in the TOP LEFT corner of each rectangle (element). Ensure you don't mix them up with other numbers (e.g. Calendar) on the page.
2) Pay attention to the filter and sort functions on the page, which, combined with scroll, can help you solve conditions like 'highest', 'cheapest', 'lowest', 'earliest', etc. Try your best to find the answer that best fits the task.

*Figure 18.* The prompt used for all zero-shot VLM agents for WebArena websites, including Claude-Sonnet-3, Claude-Sonnet-3.4, Qwen2-VL, InternVL-2.5-XComposer, LLaVa-1.6-7B, and LLaVa-1.6-34B. To be continued in Figure 19.

**Zero-Shot VLM Agent Prompt for WebArena**

* OpenStreetMap Usage Guidelines *

1) When you need to search the address of a location, you can just type the location in the 'search' bar. You don't need to use the directions button to get the address. The directions button is only used when you need to find the distance/walk/drive time between two locations.

2) When you are trying to search for a location, you may get no results. This is because the OpenStreetMap does not support approximate search. You may try to search some alternative keywords or try to find the location by yourself. Note that openstreet map does not support search phrase like 'Cafe near CMU", you should try to find it by yourself.

3) When you need to find the distance/walk/drive time between two locations, you should FIRST CLICK ON THE DIRECTIONS BUTTON (drawn as two arrows), to the right of the 'Go' Button and usually labeled as [10] or [11]. AND ONLY INPUTTING THE TWO LOCATIONS AFTER CLICKING ON THE DIRECTIONS BUTTON WHEN THE DIRECTIONS SEARCH BARS ARE SHOWN.

4) When you are trying to type some locations in the directions search bar, sometimes you may receive an alert of 'couldn't locate' followed by the location you typed. This means the location you typed is not found in the map. Do not immediately try something else. You need to quit the direction and find the precise name of this location by searching it in the map first.

5) When you search the walk/drive/bike time, make sure that you are USING THE RIGHT MODE OF TRANSPORTATION. The default mode is usually set to 'Drive'.

6) When you need to get the DD of some location, you need to click the location shown in the search result in the left part of the screen. The DD will be shown then starting with 'Location:'.

7) When you need to answer the zip code of some location, you should directly answer the 5-digit zip code. The answer shoule be "15232" instead of "The zip code of the location is 15232". Note that the zipcode will be displayed in the search result, you don't need to click the location to the information page to find the zip code.

8) When you need to answer the phone of some location, please omit the part of the country code. The answer should be "4122683259" instead of "+1 412 268 3259".

* Reddit Usage Guidelines *

1) You are already in the reddit website, though you may not see the 'reddit' in any part of the screenshot. You do not need to further navigate to the reddit website.

2) When you want to find a subreddit, you need to first navigate to Forums to see the list of subreddits. Under forums, you will see only a subset of subreddits. To get the full list of subreddits, you need to navigate to the Alphabetical option. To know you can see the full list of subreddits, you will see 'All Forums' in the observation. Often you will not find a focused subreddit that exactly matches your query. In that case, go ahead with the closest relevant subreddit. To know that you have reached a subreddit successfully, you will see '/f/subreddit_name' in the observation.

3) When you want to post forum in reddit, remember to fill up all the content, then click the button 'Create forum'. The button maybe located below out of the screenshot, you need to scroll down to find it.

4) When you want to ask or post something in a subreddit, you need to first find that subreddit and then finish the work. 5) forums and subreddits are the same thing.

Your reply should strictly follow the format:
Thought: Your brief thoughts (briefly summarize the info that will help ANSWER)
Action: One Action format you choose

Then the User will provide:
Observation: A labeled screenshot Given by User
Remember only execute one action in each step. For example, 'Action: Type [8]; CMU, Type [9] Pittsburgh' is not allowed. You should execute the action 'Type [8]; CMU' first, then 'Type [9] Pittsburgh' in the next step.

Remember to always make your answer simple and clear. For example, if you want to report the zip code of some location, always say "ANSWER; 06516" instead of "The zip code of the location is 06516".

*Figure 19.* The prompt used for all zero-shot VLM agents for WebArena websites, including Claude-Sonnet-3, Claude-Sonnet-3.4, Qwen2-VL, InternVL-2.5-XComposer, LLaVa-1.6-7B, and LLaVa-1.6-34B. Continued from Figure 18.

**Out-of-distribution Websites of WebVoyager for SFT (1/2)**
**Allrecipes:**
Simply Recipes: `https://www.simplyrecipes.com`
Food Network: `https://www.foodnetwork.com`
Taste of Home: `https://www.tasteofhome.com`
Yummly: `https://www.yummly.com`
Food.com: `https://www.food.com`

**Amazon:**
eBay: `https://www.ebay.com`
Walmart: `https://www.walmart.com`
Target: `https://www.target.com`
Best Buy: `https://www.bestbuy.com`
Alibaba: `https://www.alibaba.com`

**Apple:**
Samsung: `https://www.samsung.com`
Microsoft: `https://www.microsoft.com`
Sony: `https://www.sony.com`
Google Store: `https://store.google.com`
Dell: `https://www.dell.com`

**ArXiv:**
SSRN: `https://www.ssrn.com`
ResearchGate: `https://www.researchgate.net`
bioRxiv: `https://www.biorxiv.org`
IEEE Xplore: `https://ieeexplore.ieee.org`
PubMed: `https://pubmed.ncbi.nlm.nih.gov`

**GitHub:**
GitLab: `https://about.gitlab.com`
Bitbucket: `https://bitbucket.org`
SourceForge: `https://sourceforge.net`
Codebase: `https://www.codebasehq.com`
Gitea: `https://gitea.io`

**ESPN:**
CBS Sports: `https://www.cbssports.com`
Fox Sports: `https://www.foxsports.com`
NBC Sports: `https://www.nbcsports.com`
Bleacher Report: `https://www.bleacherreport.com`
Sky Sports: `https://www.skysports.com`

**Coursera:**
edX: `https://www.edx.org`
Udacity: `https://www.udacity.com`
Udemy: `https://www.udemy.com`
FutureLearn: `https://www.futurelearn.com`
Khan Academy: `https://www.khanacademy.org`

*Figure 20.* A list of 85 websites that we used to collect demonstration trajectories with Claude 3 Sonnet. In total 11220 trajectories were collected with different tasks. These websites were also used for testing the zeroshot generalization of PAE to out-of-distribution websites in Section 5. List continued in Figure 21.

---

**Out-of-distribution Websites of WebVoyager for SFT (2/2)**
**Cambridge Dictionary:**
Merriam-Webster: `https://www.merriam-webster.com`
Dictionary.com: `https://www.dictionary.com`
Oxford Learner's Dictionaries: `https://www.oxfordlearnersdictionaries.com`
Collins English Dictionary: `https://www.collinsdictionary.com`
YourDictionary: `https://www.yourdictionary.com`

**BBC News:**
CNN: `https://www.cnn.com`
Al Jazeera: `https://www.aljazeera.com`
Reuters: `https://www.reuters.com`
The Guardian: `https://www.theguardian.com`
NBC News: `https://www.nbcnews.com`

**Google Maps:**
Apple Maps: `https://maps.apple.com`
Bing Maps: `https://www.bing.com/maps`
MapQuest: `https://www.mapquest.com`
Waze: `https://www.waze.com`
Here WeGo: `https://wego.here.com`

**Google Search:**
Bing: `https://www.bing.com`
Yahoo Search: `https://search.yahoo.com`
DuckDuckGo: `https://duckduckgo.com`
Baidu: `https://www.baidu.com`
Yandex: `https://yandex.com`

**Hugging Face:**
OpenAI: `https://openai.com`
TensorFlow: `https://www.tensorflow.org`
PyTorch: `https://pytorch.org`
Kaggle: `https://www.kaggle.com`
SpaCy: `https://spacy.io`

**Wolfram Alpha:**
Google Scholar: `https://scholar.google.com`
Mathway: `https://www.mathway.com`
Symbolab: `https://www.symbolab.com`
Microsoft Math Solver: `https://mathsolver.microsoft.com`
Desmos: `https://www.desmos.com`

---

*Figure 21.* A list of 85 websites that we used to collect demonstration trajectories with Claude 3 Sonnet. In total 11220 trajectories were collected with different tasks. These websites were also used for testing the zeroshot generalization of PAE to out-of-distribution websites in Section **??**. List continued from Figure 21.

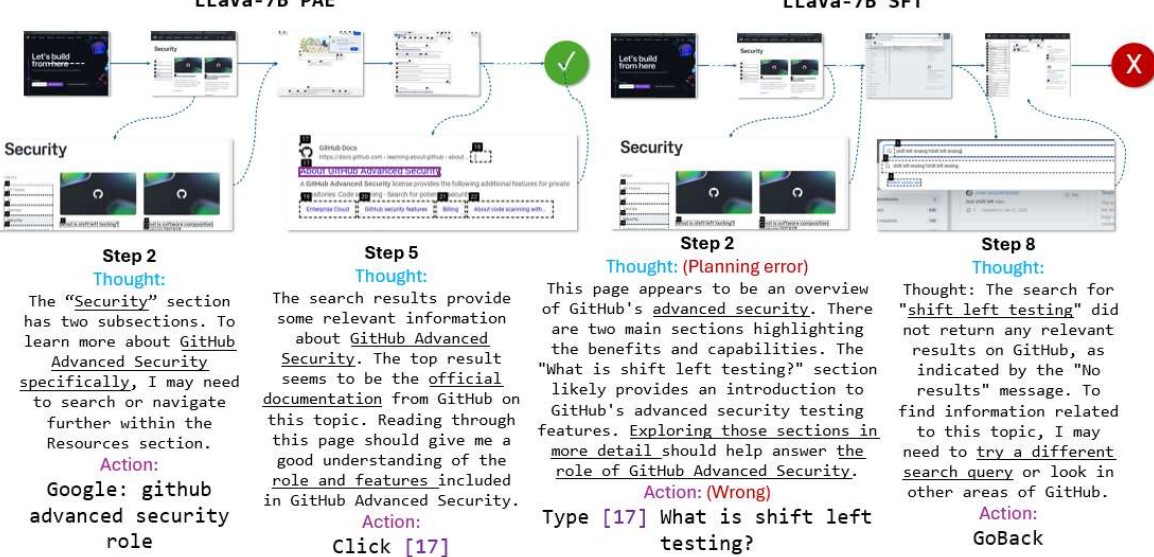

*Figure 22.* **Qualitative comparison between LLaVa-7B PAE and LLaVa-7B SFT** on the same tasks. LLaVa-7B PAE model successfully completed two tasks using learned skills from the RL training.

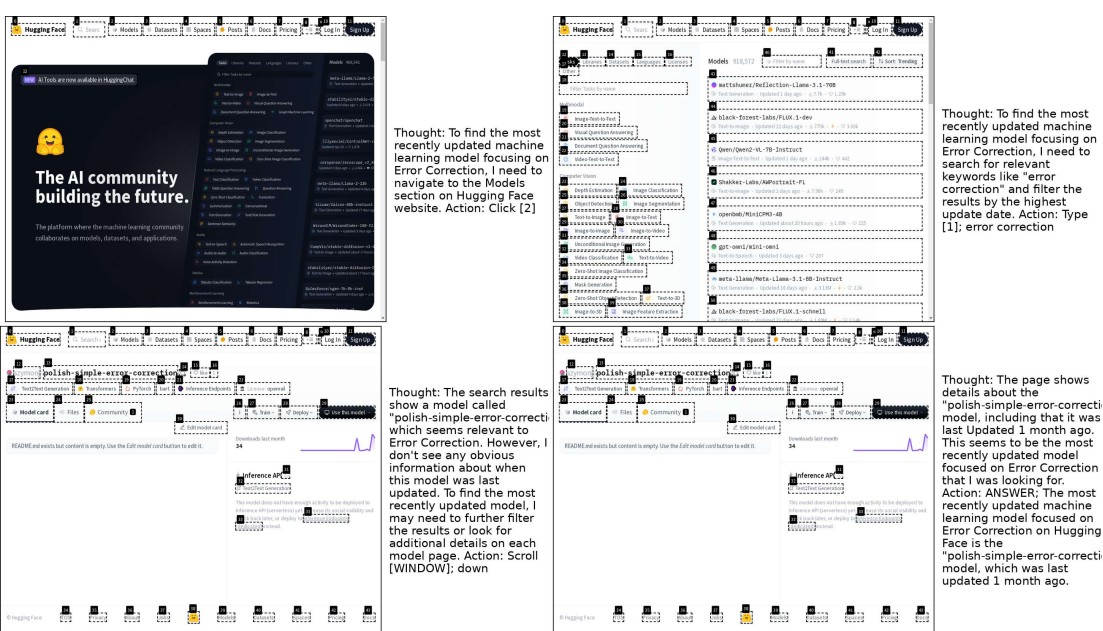

*Figure 23.* Full trajectories of success trajectory 1 in Figure 22 with task 'Find the most recently updated machine learning model on Huggingface which focuses on Error Correction' executed by model LLaVa-7B PAE.

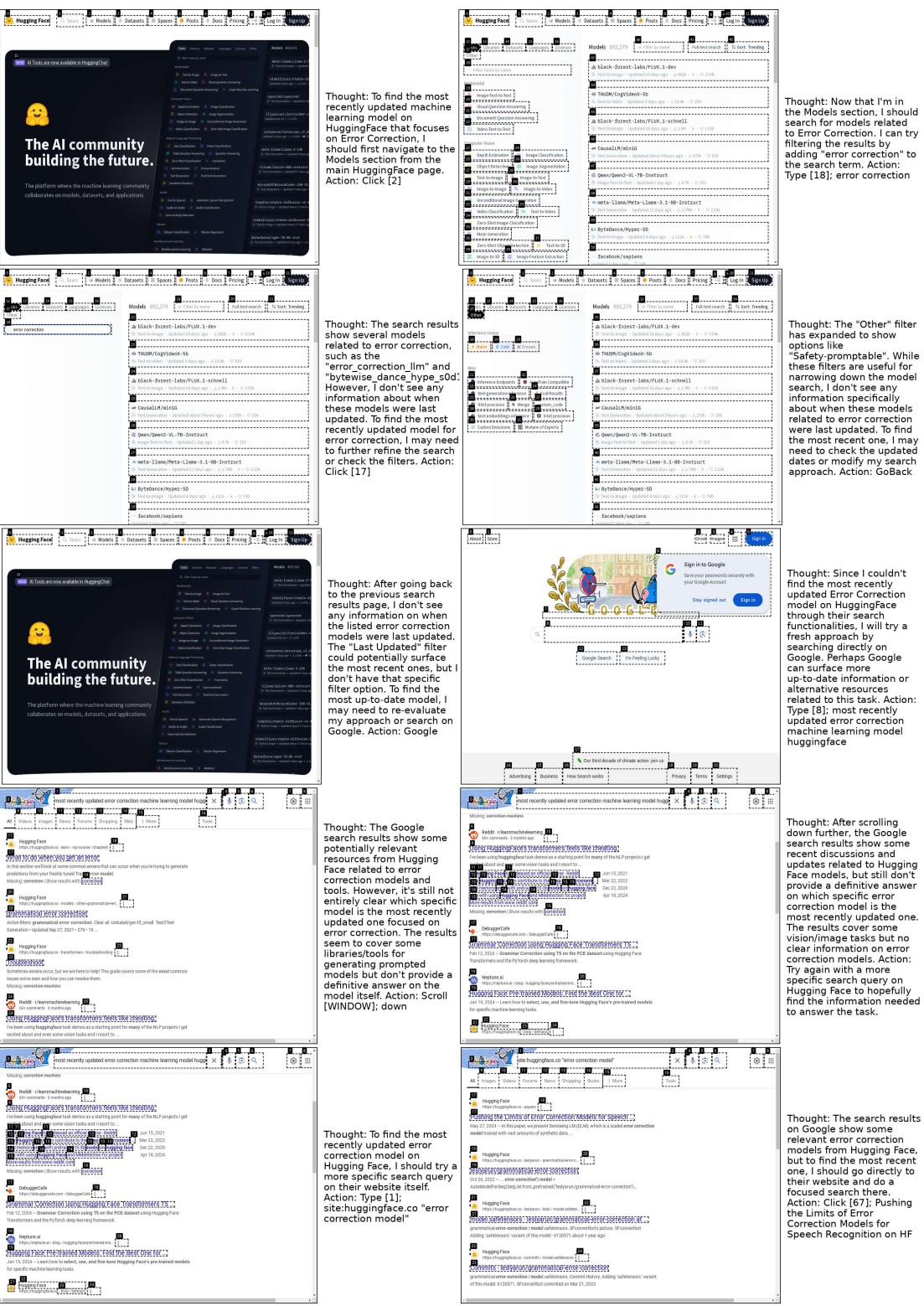

*Figure 24.* Full trajectories of fail trajectory 1 in Figure 22 with task 'Find the most recently updated machine learning model on Huggingface which focuses on Error Correction' executed by model LLaVa-7B SFT.

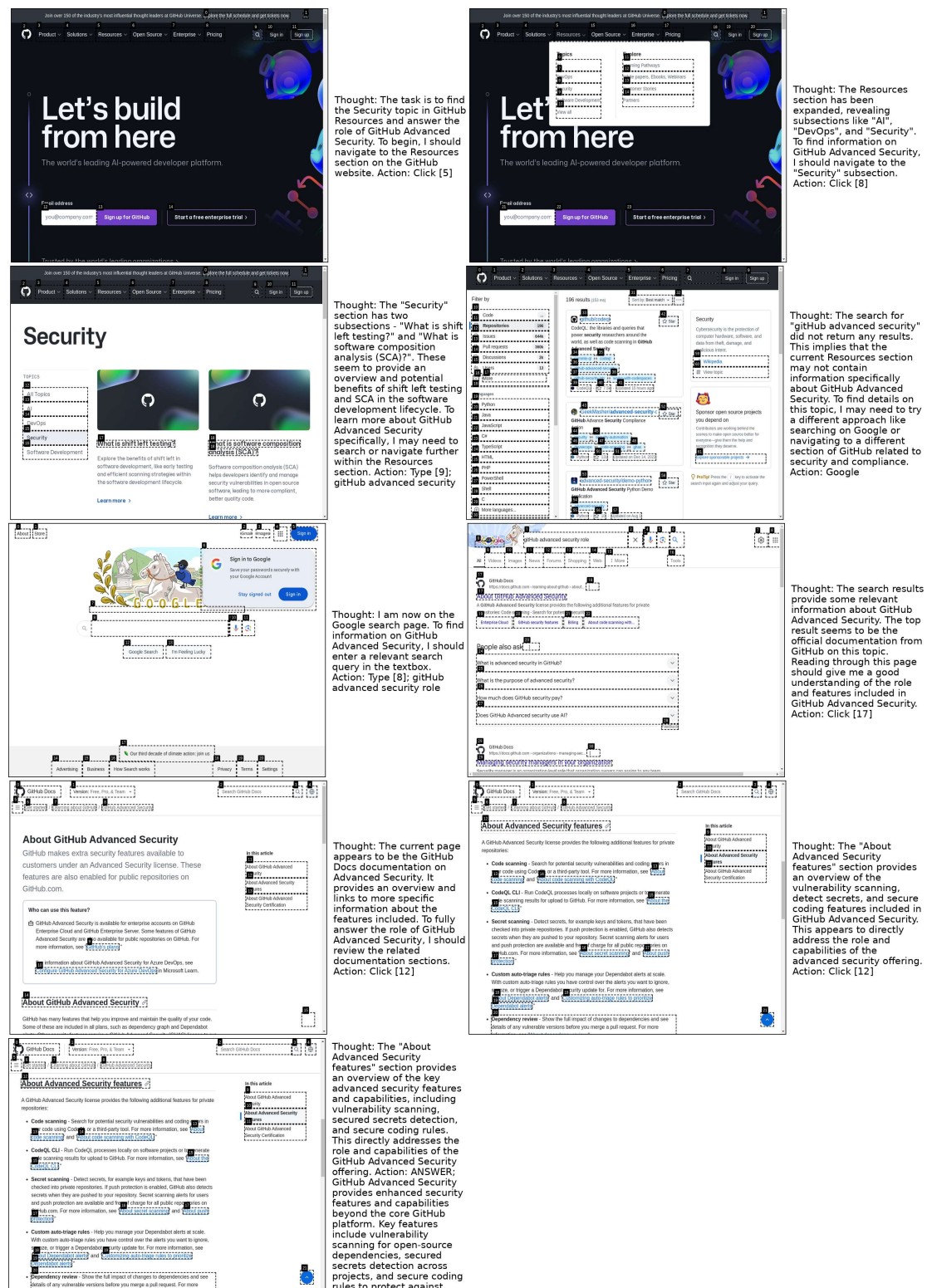

*Figure 25.* Full trajectories of success trajectory 2 in Figure 22 with task 'Find the Security topic in GitHub Resources and answer the role of GitHub Advanced Security' executed by model LLaVa-7B PAE.

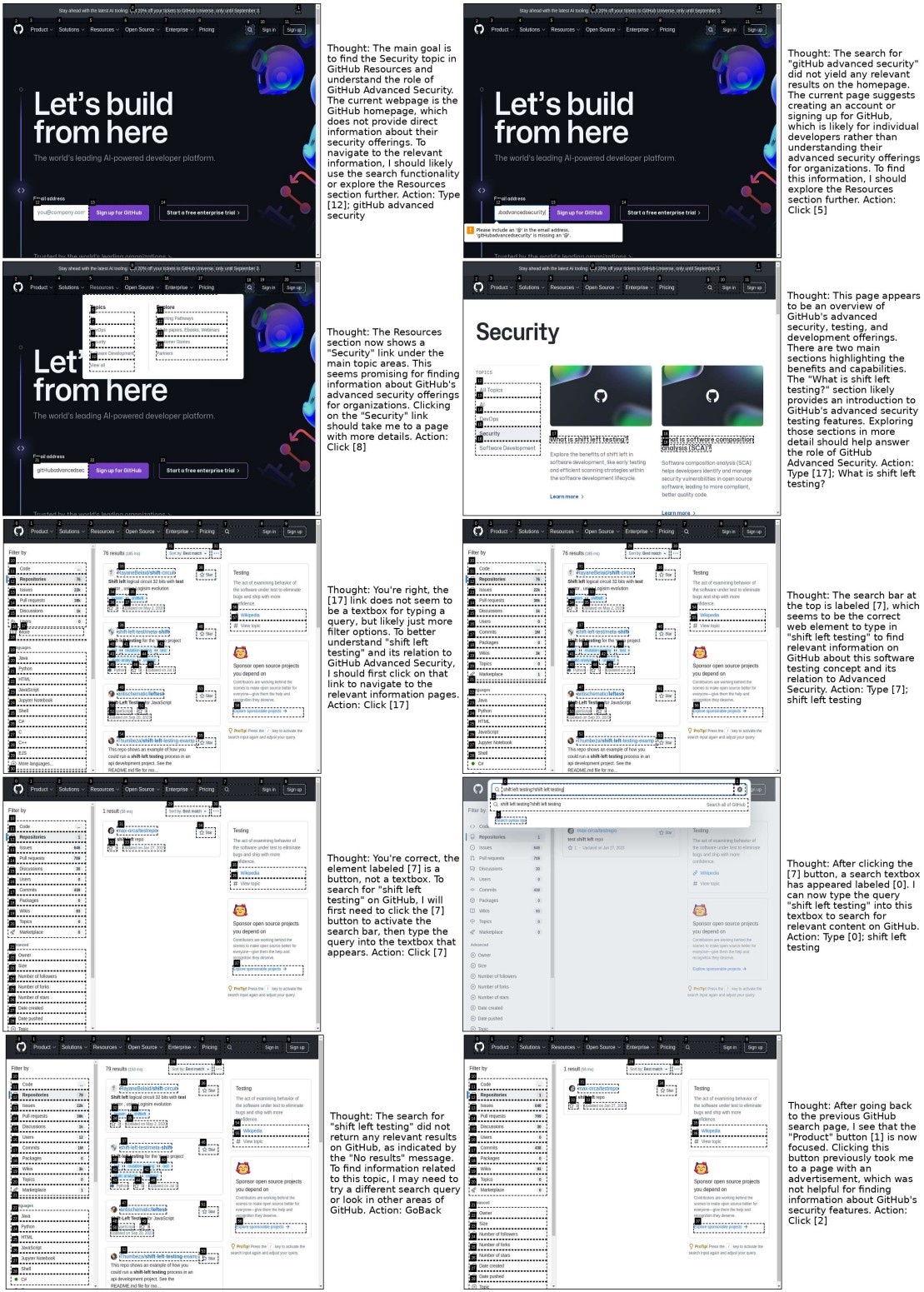

*Figure 26.* Full trajectories of fail trajectory 2 in Figure 22 with task 'Find the Security topic in GitHub Resources and answer the role of GitHub Advanced Security' executed by model LLaVa-7B SFT.

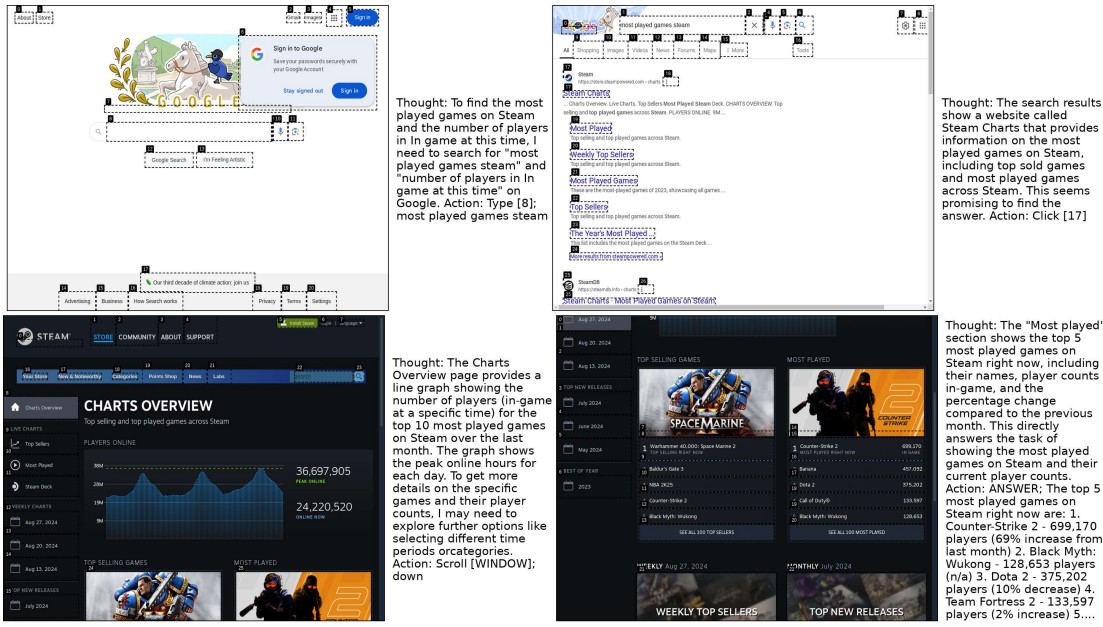

*Figure 27.* Extra full trajectories of successful trajectory 1 with task 'Show most played games in Steam. And tell me the number of players in In game at this time' executed by model LLaVa-7B PAE.

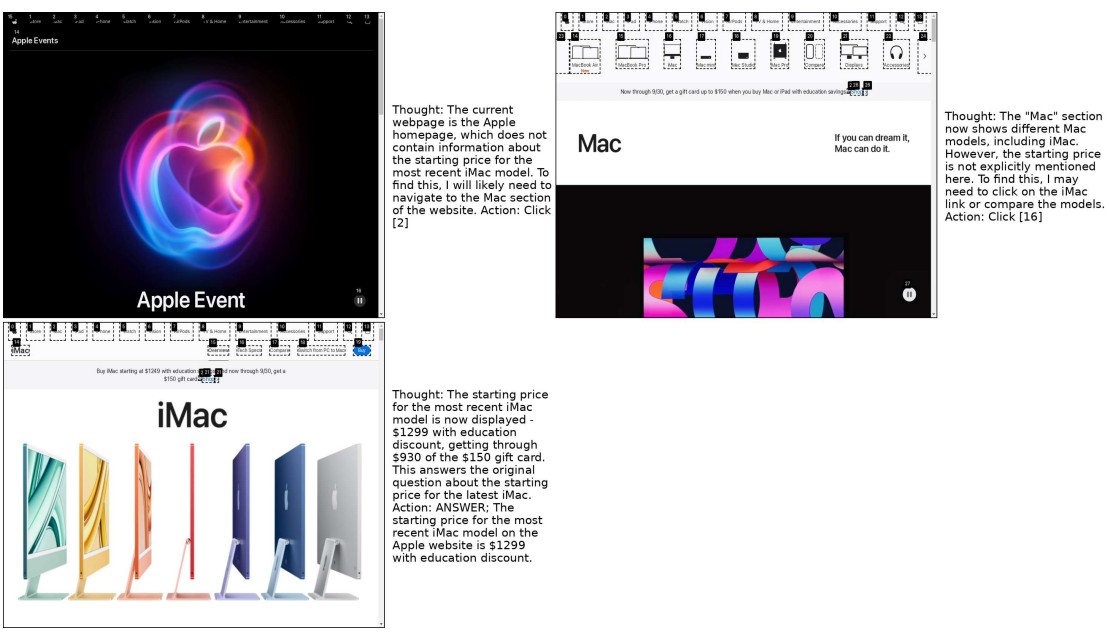

*Figure 28.* Extra full trajectories of successful trajectory 2 with task 'Find out the starting price for the most recent model of the iMac on the Apple website' executed by model LLaVa-7B PAE.

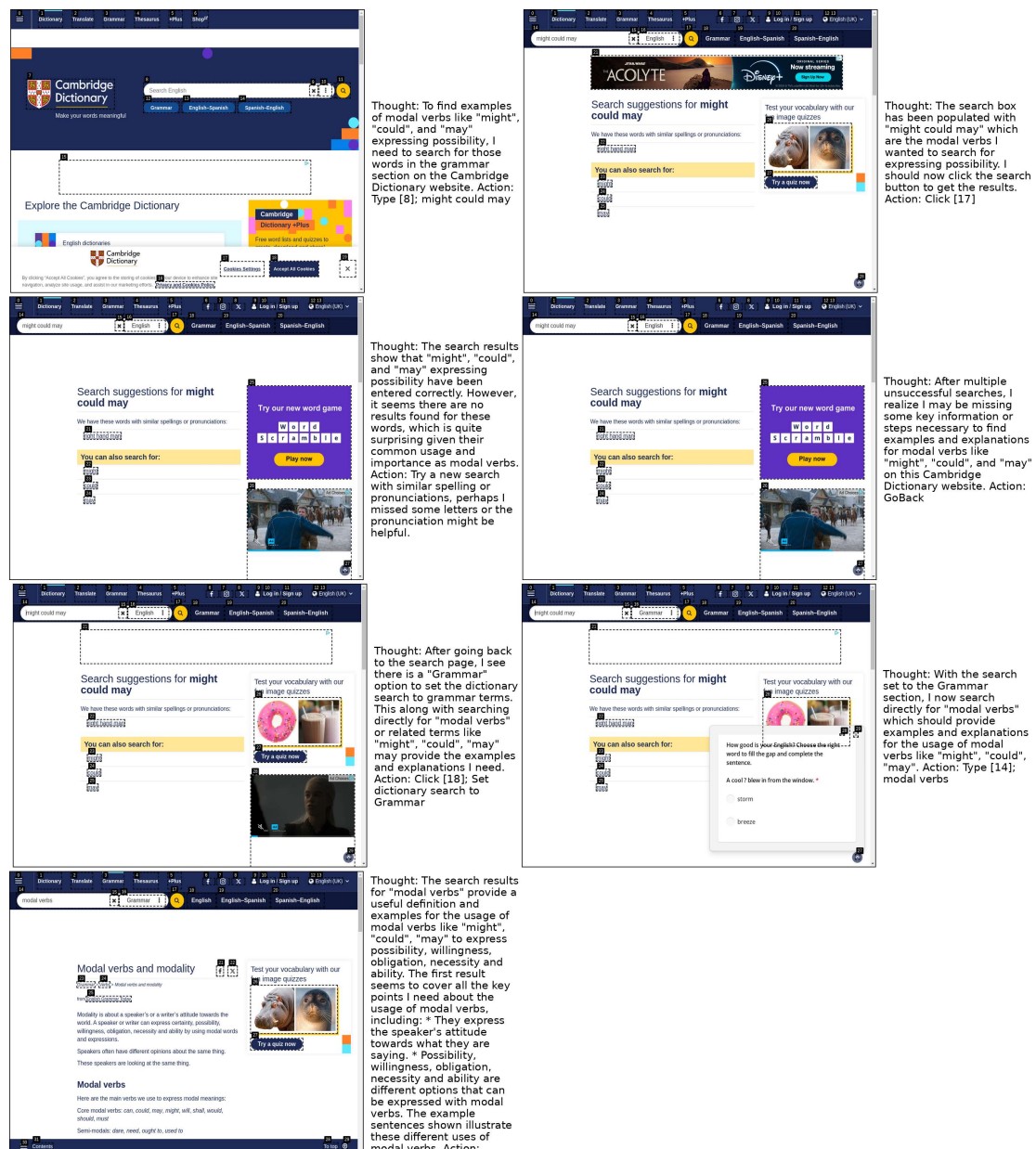

*Figure 29.* Extra full trajectories of successful trajectory 3 with task 'Look up the use of modal verbs in grammar section for expressing possibility (e.g., 'might', 'could', 'may') and find examples of their usage in sentences on the Cambridge Dictionary' executed by model LLaVa-7B PAE.

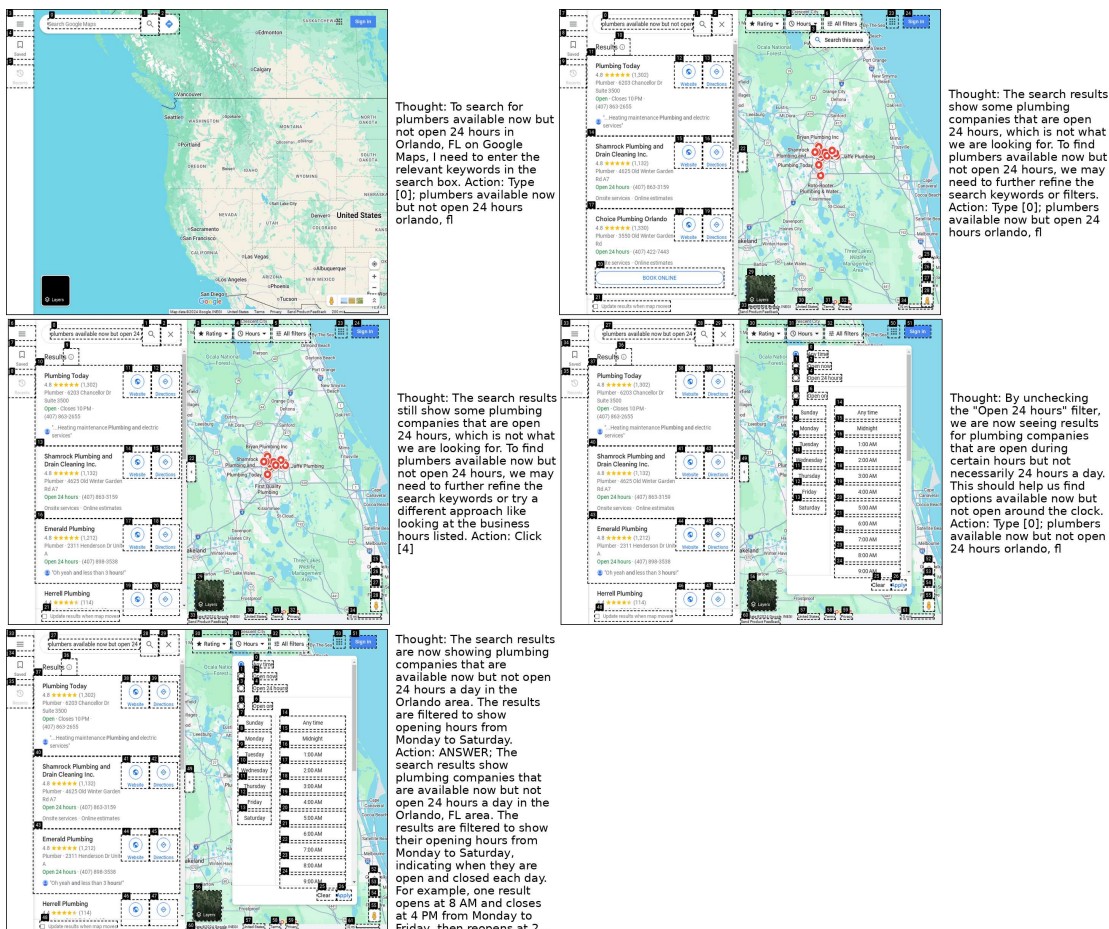

*Figure 30.* Extra full trajectories of successful trajectory 4 with task 'Search for plumbers available now but not open 24 hours in Orlando, FL' executed by model LLaVa-7B PAE.

