# OpenReview forum: "Proposer-Agent-Evaluator (PAE): Autonomous Skill Discovery For Foundation Model Internet Agents"
_ICML.cc/2025/Conference — ICML 2025 poster_

### Official Review · Reviewer_T3UV · 2025-03-12

**Overall Recommendation:** 3

**Summary:**

This paper introduces Proposer-Agent-Evaluator (PAE), a learning framework designed to enable foundation model-based Internet agents to autonomously discover and refine new skills without human supervision. PAE consists of three core components: (1) a task proposer, which generates skill acquisition tasks based on website context, (2) an agent policy, which attempts the tasks, and (3) an autonomous evaluator, which assesses success based on visual observations. The evaluation signal is then used to refine the agent’s policy through reinforcement learning. The paper demonstrates PAE’s effectiveness on web navigation benchmarks, particularly WebVoyager and WebArena, where it achieves a 50% relative improvement in success rate.

## update after rebuttal
I keep my rating since most of my concerns have been addressed. Regarding the long-term agents, I was thinking that some works—such as OpenManus—have started to support this direction. However, since these developments occurred within the past three months, they may not fall within the scope of this review’s evaluation period. Thus, I would keep my rating.

**Claims And Evidence:**

The paper claims that:

1. PAE enables autonomous skill discovery
2. PAE improves generalization
3. PAE’s improvement is not strictly dependent on stronger models for evaluation and task generation

These claims are supported by Table 1, 2, and 3.

**Essential References Not Discussed:**

None

**Experimental Designs Or Analyses:**

The experiments effectively test the proposed method by comparing different agent training strategies and task evaluation techniques. The following aspects strengthen the study:

- Comparison with SFT baselines, which highlights PAE’s advantages.
- Scaling experiments, showing consistent performance improvements across different model sizes.
- Evaluation on unseen tasks and websites, demonstrating generalization.

However, failure analysis is limited—cases where PAE-generated tasks lead to incorrect generalizations or inefficient behaviors are not thoroughly discussed.

**Methods And Evaluation Criteria:**

The task proposer utilizes contextual information (such as website names or user demonstrations) to generate training tasks. The agent policy is trained using reinforcement learning, incorporating a reasoning step before execution. The evaluator assesses task success through a binary (0/1) reward signal based on final state screenshots. Evaluation is conducted on WebVoyager and WebArena, with success rates compared against SFT-trained models and proprietary VLMs.

The methodology is well-structured, but long-term evaluation, e.g., whether PAE-discovered skills remain useful over extended training periods is not explored.

**Other Comments Or Suggestions:**

None

**Other Strengths And Weaknesses:**

Strengths:

- Addresses scalability issues in web-based agents by autonomously generating skill acquisition tasks.
- Robust across different models, showing that PAE generalizes well beyond a specific agent architecture.
- Strong empirical results, demonstrating zero-shot generalization to unseen websites.

Weaknesses:

- The paper does not discuss whether the learned skills persist over time or if they degrade when learning new ones. Understanding skill retention is crucial for real-world deployment.
- While the agent receives reward feedback, the paper does not explicitly explain how it tracks what skills it has acquired. Without an explicit tracking mechanism, the agent might relearn already mastered skills instead of focusing on truly novel capabilities.
- Sparse reward signal—the 0/1 evaluation approach might miss intermediate learning signals that could improve policy refinement.

**Questions For Authors:**

1. How does PAE handle situations where the proposed task is infeasible or ambiguous? Would the system benefit from a self-correction mechanism?
2. Does the agent recognize when to reuse a learned skill versus learning from scratch?
3. Does PAE facilitate long-term skill retention? Do discovered skills remain useful across multiple training phases, or do they degrade over time due to task shifts?
4. Would intermediate reward signals (e.g., step-based evaluations) improve agent learning? The paper suggests outcome-based evaluation is more stable, but step-based signals could provide earlier correction if the agent starts diverging from the goal.

**Relation To Broader Scientific Literature:**

The paper contribute to self-supervised skill discovery with RL and web-based foundation model agents.

**Theoretical Claims:**

The paper does not proposed theoretical claims.

---

> ### Author Rebuttal · Authors · 2025-04-01
>
> Thank you for your review and feedback on the paper. We have provided additional clarifications to the questions that you have raised on how PAE handles cases where the proposed task is infeasible, how the agent recognizes when to reuse a learned skill, and whether intermediate reward signals can help. **Please let us know if your concerns are addressed and if so, we would appreciate it if you could re-evaluate our work based on this context. We are happy to discuss further.**
>
>
> ## However, failure analysis is limited—cases where PAE-generated tasks lead to incorrect generalizations or inefficient behaviors are not thoroughly discussed.
> We actually **do have comprehensive analysis in appendix B with manual classifications of the trajectories from each model**. We have also provided qualitative examples of each failure mode in Figures 7-11. We will include them in a revised version of the paper.
>
>
> ## How does PAE handle the situations where the proposed task is infeasible or ambiguous?
> An important factor for PAE to choose to use vanilla REINFORCE without a baseline value function is that this loss function (line 239 in the manuscript) automatically takes care of the cases when the proposed task is infeasible or ambiguous. In these cases, the agent would not be able to finish the task and therefore can only get a reward 0. As shown in the loss function in line 239, if the trajectory reward is 0, it automatically zeros out the loss so it does not affect the training process, except a slight waste of sample efficiency during online trajectory collection.
>
>
>
>
> ## Does the agent recognize when to reuse a learned skill versus learning from scratch? How does PAE facilitate long-term skill retention?
>
> A crucial difference in skill discovery considered in this setting of web agents versus skill discovery in traditional hierarchical deep reinforcement learning [1, 2] is that most tasks that current sota open-source VLM web agents can complete are rather atomic and short-horizon (~10 steps), such as “Find the Easy Vegetarian Spinach Lasagna recipe on Allrecipes and tell me what the latest review says” as shown in appendix Figure 7. Under such a setting, skills are essentially equivalent to the tasks that the agents are practicing on so the issues of identifying new skills and long-term skill retention are not major concerns at this stage. However, as open-source models get more capable and are able to handle more complicated tasks, we believe that there will be more interesting research problems related to those concerns and the framework of PAE will serve as a foundation to facilitate such research.
>
> [1] HIQL: Offline Goal-Conditioned RL with Latent States as Actions
>
> [2] Data-Efficient Hierarchical Reinforcement Learning
>
>
>
> ## Would intermediate reward signals improve agent learning?
> The choice of the reward is actually an important ablation choice that we have made to maximize the exploitation of asymmetric capabilities of SOTA VLMs as skill proposers/evaluators and as agents. As shown in Figure 3, we found the use of sparse outcome rewards only achieved the best performance while other choices such as autonomous step rewards and functional evaluators tend to result in lower performances. Through a more careful inspection, we found that in many cases it would be very hard to give an accurate step-level reward because there can be multiple paths to solving the tasks. That said, we also believe that as the tasks get harder and models get more capable, how to exploit autonomous intermediate reward signals would be an important future direction to be explored.

---

> > ### Comment · Reviewer_T3UV · 2025-04-02
> >
> > Thank you for the clarification. Most of my concerns have been addressed. Regarding the long-term agents, I was thinking that some works—such as OpenManus—have started to support this direction. However, since these developments occurred within the past three months, they may not fall within the scope of this review’s evaluation period. Thus, I would remain my rating.

---

> > > ### Author Response · Authors · 2025-04-08
> > >
> > > Thank you so much for your response and appreciation of our response!

---

### Official Review · Reviewer_D3jd · 2025-03-12

**Overall Recommendation:** 3

**Summary:**

The paper presents a novel framework, Proposer-Agent-Evaluator (PAE), enabling foundation-model internet agents to autonomously discover and learn diverse skills without relying on human-annotated task templates. By leveraging visual-language models (VLMs), PAE autonomously generates tasks (Proposer), executes them using an agent policy guided by chain-of-thought reasoning, and autonomously evaluates task success. Experiments demonstrate significant improvements in zero-shot generalization performance across realistic internet browsing tasks compared to existing baselines.

**Claims And Evidence:**

Yes

**Essential References Not Discussed:**

N/A

**Experimental Designs Or Analyses:**

Yes

**Methods And Evaluation Criteria:**

Yes

**Other Comments Or Suggestions:**

N/A

**Other Strengths And Weaknesses:**

Strengths:
1. The paper explores an interesting and practical problem—how AI models can automatically discover useful skills without relying on humans manually labeling tasks, which could greatly expand what these models can do in the real world.
2. The proposed framework (PAE) is straightforward and well-designed. It enables the model to come up with tasks by itself, figure out how to solve them, and evaluate its own performance, without needing human input every step of the way.
3. The authors ran thorough experiments, clearly showing that their method outperforms existing approaches and achieves strong results.

Weaknesses:
1. The paper lacks comprehensive comparisons with alternative reinforcement learning algorithms and does not sufficiently explore performance differences with other large language models.
2. The paper conducts experiments only on WebVoyager and WebArena environments. Although these environments are rich, the tasks tested are predominantly simple, single-turn interactions (e.g., searching products or clicking elements). It would be beneficial to experiment with multi-turn interactive tasks or other long-term interaction scenarios.
3. The paper lacks a detailed analysis of computational efficiency, including specifics on computational resources, data efficiency, and training costs.

**Questions For Authors:**

N/A

**Relation To Broader Scientific Literature:**

N/A

**Theoretical Claims:**

Yes

---

> ### Author Rebuttal · Authors · 2025-04-01
>
> Thank you for your review and feedback on the paper. We have included additional clarifications in the multi-turn nature of WebArena and WebVoyager, and details of computational efficiency. We are running additional experiments comparing different RL algorithms and we will share them here once we have preliminary conclusions. **Please let us know if your concerns are addressed and if so, we would appreciate it if you could re-evaluate our work based on this context. We are happy to discuss further.**
>
>
>
> ## Experiments are only conducted on single-turn interactions on WebVoyager and WebArena. It would be beneficial to experiment with multi-turn interactive tasks or long-term interaction scenarios
>
> To the best of our understanding, WebVoyager and WebArena environments are multi-turn and interactive environments where the agent needs to interact with the websites for a few steps (~10-20 steps) to be able to complete a task like “Find the Easy Vegetarian Spinach Lasagna recipe on Allrecipes and tell me what the latest review says” as shown in appendix Figure 7. While we agree that these environments are still relatively short-horizon in that most tasks require between 10-20 steps for successful executions, these are already the hardest interactive benchmarks on which existing open-source VLM web agents can achieve non-trivial performance results. We expect that more research in credit assignments over longer horizons would be possible in the future as the capabilities of open-source VLMs improve.
>
> ## The paper lacks a detailed analysis of computational efficiency
> Our main experiments of 7B models were conducted using 1 8x40G A100 and 3 8x24G A10G machines where the gradient update steps are performed on A100 only and A10G machines are only used for distributed data collection (i.e. trajectory rollouts). The trajectory collection phase takes around 90% of the training time while gradient updates only take 10%. The trajectory collection speed using all 4 machines is around 1k-2k trajectories per hour, so 7B experiments can be completed within 2 days. Our run of 34B model was conducted using 3 8x40G A100, where gradient update steps are performed on one of the A100 machines and all machines participate in trajectory collections. The training for the 34B model takes around 5 days under this setup. We will include this section in the appendix for the revised version.
>
>
> ## The paper lacks comprehensive comparisons with other reinforcement learning algorithms and does not sufficiently explore performance differences with other LLMs.
>
> The novel finding of this paper is to study whether an entire self-improvement framework of LLMs for web agents is possible and what the necessary design choices are. While additional RL results are good to have, because of the computational constraints discussed in the last response, we were unable to perform systematic tuning for other reinforcement learning algorithms like ppo and grpo with more hyperparameters and more sample complexity. Therefore, we refrained from drawing comparisons between different RL algorithms. With that being said, we are running additional experiments with PPO as the RL algorithm.  We will share the results here once we have preliminary conclusions.

---

### Official Review · Reviewer_FPLC · 2025-03-14

**Overall Recommendation:** 2

**Summary:**

The authors propose Proposer-Agent-Evaluator (PAE) as a web agent that generates probable tasks (instructions) by itself and performs RL fine-tuning with binary rewards from VLM-based evaluators. For the generation of tasks for training, they employ a number of different pieces of information about target websites, such as website names and human demonstrations. Using the Set-of-Marks (SoM) annotations for web page screenshots, they fine-tune their VLM with binary rewards that are generated by prompt-based VLM evaluators, which take the last three web page screenshots and the agent's answer as input. The authors suggest that their approach brings performance improvements relative to supervised fine-tuned VLM agents.

## update after rebuttal

Thank you for providing the rebuttal.

Regarding the originality of this work, using an autonomous evaluation and fine-tuning based on the filtering was a practice used by "Autonomous Evaluation and Refinement of Digital Agents" (Pan et al., 2024, cited by this submission, as well), although the filtered behavior cloning was for their iOS agents, and it was first presented as a preprint on Apr 9, 2024 and as a non-preprint at the Multimodal Algorithmic Reasoning Workshop (CVPR) on Jun 17, 2024. Using foundation models for task proposal in the web navigation domain has been used by WebVoyager (He et al., ACL 2024) or even Mind2Web (Deng et al., NeurIPS 2023). As I mentioned in my original review, I am not suggesting that there are no differences from existing work, and I agree that empirically showing that self-improvement can work could be meaningful. But overall, my view regarding the originality remains similar.

For the empirical evaluation, I believe demonstrating the effectiveness of the proposed framework with a stronger setup on a more difficult set of tasks would provide better empirical takeaways.

While I appreciate the authors for providing the response to my review, I keep my original rating due to my primary concerns about this work.

**Claims And Evidence:**

- The assumption about providing the knowledge and information about desired websites to the task proposer is reasonable, because coming up with tasks that are useful and executable at the same time for different websites can be challenging.
- On the other hand, the use of demonstrations on the websites comes with its own challenge of being outdated as the websites get updated. This can make the proposed approach less "automated," especially for websites that are not very well-known or get frequently updated. In this sense, the demonstration (screenshot)-guided task generation may not fully solve the issue of anticipating what is possible on each website.

**Essential References Not Discussed:**

I think relevant papers are reasonably cited.

**Experimental Designs Or Analyses:**

- Although the authors provide an explanation for creating the "easy" split of WebArena, I believe using the original WebArena (even if it does not include the full list of websites it provides) would be better for comparison across different papers.

**Methods And Evaluation Criteria:**

- The proposed method, Proposer-Agent-Evaluator, makes sense for web agents. As mentioned in the Claims and Evidence section, generating tasks in real-world domains like web navigation should be well-grounded.

**Other Comments Or Suggestions:**

- I think the readability and presentation of the manuscript can be improved. For instance, many of the tables and figures have too small fonts and are hard to read.

**Other Strengths And Weaknesses:**

- Overall, the originality of this work looks somewhat limited to me. My understanding is that using foundation models to generate task instructions, trajectory data collection with agents, and foundation model-based evaluators for evaluation and trajectory filtering (some of the corresponding prior work for these items is already mentioned in this work) are part of the standard practice for training web agents, these days. While some decisions for these components may have difference, I think the overall pipeline is similar to the norm.
- The use of the term "skill" may not be appropriate in this context and may be replaced with a clearer term. For instance, I don't think there would be much difference if "policy" was used, instead. At least in the context of agents and reinforcement learning, "skill" and "skill discovery" sound more appropriate for focusing on learning behaviors at levels smaller/lower than the task level (i.e., "skill" level), so that they can be combined or leveraged to solve more complex downstream tasks.

**Questions For Authors:**

N/A

**Relation To Broader Scientific Literature:**

- The task proposer may be applicable in other domains or environments. In that sense, there is a possibility of broader application of this work.

**Theoretical Claims:**

There is not much of theoretical claims from this submission.

---

> ### Author Rebuttal · Authors · 2025-04-01
>
> Thank you for your review and feedback on the paper. Just a gentle reminder that by the definition established by the ICML2025 committee, concurrent work includes all papers from within 4 months of the submission deadline. This applies to all other pipelines for web agents [1,2,3] with similar components (task proposers, agents, and evaluators). As such, they should not be considered as an established standard. With that being said, we also provide additional discussions on the differences with these works to address your concerns regarding novelty and also answer your other concerns below. **Please let us know if your concerns are addressed and if so, we would appreciate it if you could re-evaluate our work based on this context. We are happy to discuss further.**
>
> ## Overall, the originality of this work looks somewhat limited to me
> First, we would like to emphasize that while other pipelines [1,2,3] with similar components (task proposers, agents, and evaluators) have been applied to web agents because of the effectiveness of these components, they are all made public within 4 months of the submission deadline of ICML. As stated in the reviewer guideline of ICML, they should be considered as concurrent instead of an established standard and “Authors cannot expect to discuss other papers that have only been made publicly available within four months of the submission deadline. “.
>
> Additionally, a key difference is that [1,2,3] generate instructions and evaluation rewards from stronger proprietary models such as GPT4o to train weaker models, while our PAE system focuses on exploiting the asymmetric capabilities of foundation model web agents to achieve self-improvements, where even weaker models can be used to improve the performance of stronger agents. We propose that this is a meaningful contribution, as it demonstrates a path to advance state of the art capabilities. In our main results, we have also performed extensive experiment analysis seeking to understand when such self-improvements are possible (the use of a reasoning step, the use of sparse outcome-based reward, different choices of the models etc) and how well such self-improvements can generalize (to unseen tasks and unseen websites). This is the novel finding of this paper.
>
>
> [1] Nnetscape navigator: Complex demonstrations for web agents without a demonstrator, 2024.
> [2] Openwebvoyager: Building multimodal web agents via iterative real-world exploration, feedback and optimization, 2024b.
> [3] Webrl: Training llm web agents via self-evolving online curriculum reinforcement learning, 2024.
>
>
> ## Using the original WebArena would give better comparisons across different papers.
> In addition to experiments on WebArena Easy as reported in the main text, in appendix G Table 4 we also reported the performance of our models on the original WebArena using only screenshots as observations, and provided additional explanations in terms of why WebArena Easy is used as opposed to original WebArena. For your convenience, we also include results from Table 4 below. These results in Table 4 would be directly comparable with other papers testing screenshot-only models on WebArena. At the time when the main experiments of this paper were conducted, the best open-source VLM that we were able to train was LLaVa-Next but they were not able to achieve higher than chance performance (around 5%) for most of the websites on WebArena. In these cases, RL would be almost impossible because the base model does not perform meaningful explorations for these hard tasks.
>
> |                |                 | OpenStreetMap | PostMill | OneStopMarket | Average |
> |----------------|-----------------|--------------:|---------:|--------------:|--------:|
> | Proprietary    | Claude 3 Sonnet|           24.3|      10.6|          11.2 |    14.6 |
> | Open-Source    | Qwen2VL-7B     |            0.7|       0.0|           1.3 |     0.7 |
> | Open-Source          | InternVL2.5-8B |            2.6|       0.2|           3.3 |     2.3 |
> | Open-Source          | LLaVa-7B       |            0.0|       0.0|           0.0 |     0.0 |
> | Ours           | LLaVa-7B SFT   |           15.2|       1.4|           5.8 |     7.2 |
>
>
>
> ## The use of the term "skill" may not be appropriate in this context and may be replaced with a clearer term.
> We used the term “skill” because most of the tasks that can be completed by open-source VLM web agents were restricted to rather short-horizon and atomic tasks like “find a chicken recipe with more than 4.8 reviews”. To avoid confusion, we will replace them with the term “policy” as suggested in the context.
>
> ## The readability and presentation of the manuscript can be improved. Many of the tables and figures have too small fonts to read.
> Thank you for the concrete advice on improving the presentation of the paper. We will increase the fonts in the updated version.

---

> > ### Comment · Reviewer_FPLC · 2025-04-04
> >
> > Thank you for providing the rebuttal.
> >
> > Regarding the originality of this work, using an autonomous evaluation and fine-tuning based on the filtering was a practice used by "Autonomous Evaluation and Refinement of Digital Agents" (Pan et al., 2024, cited by this submission, as well), although the filtered behavior cloning was for their iOS agents, and it was first presented as a preprint on Apr 9, 2024 and as a non-preprint at the Multimodal Algorithmic Reasoning Workshop (CVPR) on Jun 17, 2024. Using foundation models for task proposal in the web navigation domain has been used by WebVoyager (He et al., ACL 2024) or even Mind2Web (Deng et al., NeurIPS 2023).
> > As I mentioned in my original review, I am not suggesting that there are no differences from existing work, and I agree that empirically showing that self-improvement can work could be meaningful. But overall, my view regarding the originality remains similar.
> >
> > For the empirical evaluation, I believe demonstrating the effectiveness of the proposed framework with a stronger setup on a more difficult set of tasks would provide better empirical takeaways.
> >
> > While I appreciate the authors for providing the response to my review, I keep my original rating due to my primary concerns about this work.

---

> > > ### Author Response · Authors · 2025-04-08
> > >
> > > We appreciate your response to our rebuttal, and thank you for acknowledging the difference of our paper with existing work and the meaningful contribution of showing that self-improvement can work. We agree that carefully ablating the design choices of each component in the system and performing extensive experiments to understand self-improvement are exactly the contributions of this paper, instead of an individual novel component.
> > >
> > > We understand that you suggest we try the proposed framework on more difficult set of tasks but WebVoyager and WebArena are the hardest multi-turn web agent benchmark at the time when the experiments were conducted (as shown in Table 1 and Table 2, even the strongest Claude Sonnet 3.5 can only achieve ~50% success rate). We have also tried a harder set of tasks on WebArena in Table 4 but unfortunately no open-source VLMs at the time of experiments (even after SFT) can perform meaningful explorations on it, so the tasks considered in our paper are already the hardest multi-turn web agent possible for open-source VLM web agent.

---

### Decision · Program_Chairs · 2025-05-01

**Decision:**

Accept (poster)

**Comment:**

This paper proposes PAE (Propose, Act, Evaluate), a self-improvement framework for training web agents using only open-source vision-language models (VLMs). The system leverages asymmetric model capabilities by using weaker models to propose tasks and evaluate outcomes, thereby improving stronger agents through reinforcement learning without human supervision or proprietary models. The framework is evaluated on WebArena and WebVoyager, two of the most challenging multi-turn web interaction benchmarks, and includes extensive ablations to analyze the effect of different design choices such as reward structure, model selection, and reasoning steps.

The paper is well-motivated and introduces a scalable, practical approach for self-improving web agents. Its main strength lies in the system-level insight that weaker models can guide the improvement of stronger ones, which is a novel and impactful perspective. The experimental analysis is thorough, covering design choices and ablations across challenging benchmarks. However, the work lacks extensive comparison with alternative RL algorithms and baseline models, and long-term skill retention remains unexplored. While some components mirror recent concurrent works, the authors convincingly clarify their contributions and novelty under ICML’s concurrent work guidelines.

In sum, I slightly lean toward accepting this paper, depending on if there is room in the program, and I would not mind if my recommendation was bumped down.